# A multifunctional smart field-programmable radio frequency surface

Tianzhi Li [1] ✉, Yang Yu [1], Yutong Liu [1], Ranjith R. Unnithan [1], Ryan J. McDermott [1], Dominique Schreurs [2], Robin Evans[1] & Efstratios Skafidas [1] ✉

Antennas that can operate across multiple communication standards have remained a challenge. To address these limitations, we propose a Field-Programmable Radio Frequency Surface (FPRFS), which is based on manipulating current flow on its surface to achieve desirable RF characteristics. In this work, we demonstrate that substantial enhancements in radiation efficiency can be achieved while preserving the high reconfigurability of antenna structures implemented on the FPRFS. This is accomplished by utilizing an asymmetric excitation, directing the excitation to the low-loss contiguous surface, and dynamically manipulating the imaged return current on a segmented ground plane by switches. This important insight allows for adaptable antenna performance that weakly depends on the number of RF switches or their loss. We experimentally validate that FPRFS antennas can achieve efficiencies comparable to traditionally implemented antenna counterparts. This permits the FPRFS to be effectively utilized as a productive antenna and impedance-matching network with real-time reconfigurability.

The ability to communicate without wires has been transformative and enabled many applications that were unimaginable only a few years ago[1]. With the entrenchment of wireless technologies, new challenges have arisen, including the communication spectrum becoming increasingly crowded[2] and consumer demands to implement multiple communication standards on a single handheld device[3,4]. It is not uncommon for today's devices to require the implementation of long-term evolution (LTE), 5G, Bluetooth, dual-band WiFi, and near-field communication (NFC) in a single handheld[5]. Fortunately, novel electronic Radio Frequency (RF) circuit architectures have addressed some of the challenges required to build single wideband and highly flexible transceivers[6,7]. Active circuit implementations have replaced many off-chip passive components, e.g., baluns and transmission lines[8]. These active circuits have broken the interdependency of the operating frequency to the device's geometric size. Unfortunately, achieving antennas that can implement multiple communication standards operating across many gigahertz of bandwidth has remained a

challenge, as the antenna's impedance, bandwidth, radiation pattern, and efficiency are all highly interdependent and interlinked with its geometry and size[9,10]. Antennas with reconfigurability have been an area of intense research[11–19], as they enable desirable operational flexibility that can more efficiently utilize the limited radio spectrum[20] and also offer the ability to cope with complicated RF environments[21].

Notwithstanding the significant advances, most reconfigurable antennas available today are limited in their bandwidth and their ability to adapt their input impedance and radiation pattern in response to operational requirements or changes to the RF propagation environment, such as the presence of water and ice on the antenna surface or a human body near the antenna during device operation, as shown in Fig. 1a, b. Today's reconfigurable antennas have limitations such as slow manual reconfiguration speed, limited programmability, and inefficient and lossy radiators. We propose, implement, and validate an RF structure known as a Field-Programmable RF Surface (FPRFS) to contend with these limitations. A conceptual illustration of

[1]Department of Electrical and Electronic Engineering, Faculty of Engineering and Information Technology, The University of Melbourne, Parkville, VIC 3010, Australia. [2]Div. WAVECORE, Department of Electrical Engineering (ESAT), KU Leuven, Kasteelpark Arenberg 10, 3001 Leuven, Belgium. ✉e-mail: tianzhi.li@unimelb.edu.au; sskaf@unimelb.edu.au

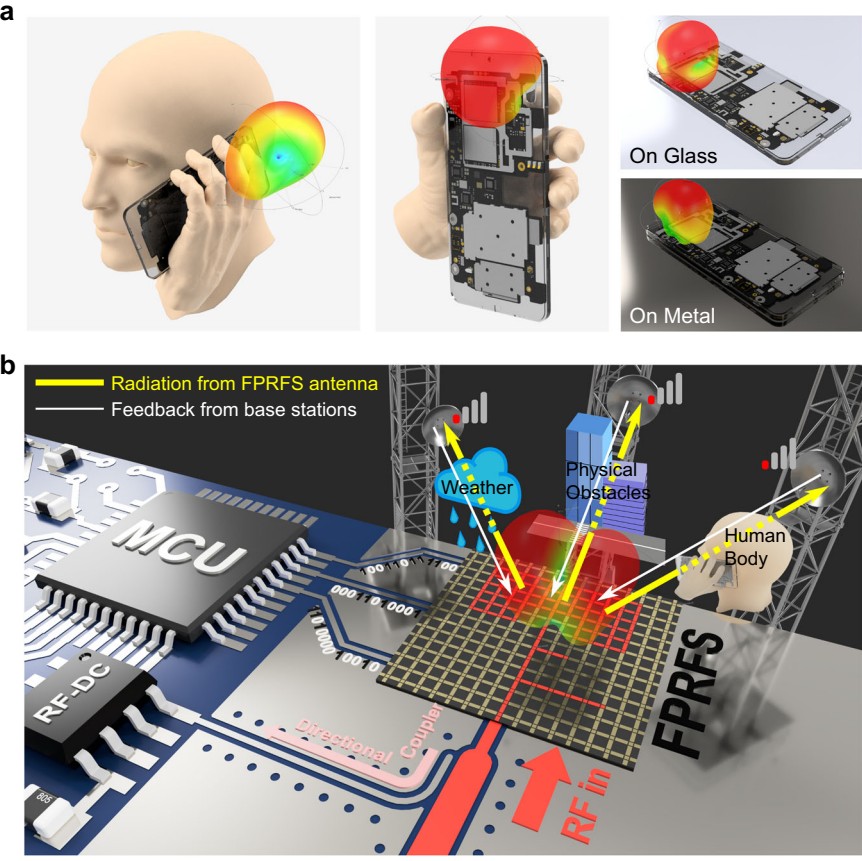

**Fig. 1 | Conceptual illustration of the smart FPRFS. a** Illustration of the varying radiation patterns of a conventional handheld device antenna when placed near the ear, held in the hand, placed on a glass surface, and placed on a metal surface. **b** Conceptual illustration of a digital smart FPRFS operating within a compact electronic device, designed to mitigate environmental factors such as rainwater, physical obstacles, and the human body. The red-highlighted pattern on the FPRFS denotes the real-time active RF conductive path, forming an arbitrary antenna and impedance-matching network topology based on signal strength feedback from remote base stations and a local coupler.

the FPRFS in a practical scenario is shown in Fig. 1b. The fundamental concept underpinning this surface is that the flow of current on this surface can be reconfigured to meet desired RF requirements. Controlling the flow of current on the surface results in a dynamic antenna that achieves desired propagation characteristics and corresponding impedance-matching capability.

This FPRFS concept is inspired by Field-Programmable Gate Arrays (FPGAs), where pre-fabricated logic gates and flexibly programmable interconnects are used to implement complex circuits[22], as well as the pixelated antenna concept[23,24]. In our proposed FPRFS, infinitesimal current pixels, that can be mathematically modeled as dipoles, are the fundamental building blocks. The small rectangular pixels can be placed in series or parallel to achieve the desired current distribution on the surface. This concept builds upon a familiar concept introduced in elementary antenna theory texts, where the properties of an electrically large antenna are derived from the integration over the spatial distribution of infinitesimal dipoles[25]. This approach is also similar to the infinitesimal dipole model (IDM) method[26,27], which has recently been proposed to model and obtain insights into complex antenna structures. Switches are commonly used to control the flow of current. Unfortunately, RF switches have high losses, and any benefit gained from configuration flexibility is heavily offset by high loss and poor radiation efficiency[28]. We explore an asymmetric excitation scheme where the current is injected into the low-loss contiguous plane to overcome the significant power loss and poor efficiency. The current flow reconfigurability is achieved by controlling the imaged return current path on the segmented RF surface, serving as the ground plane. This unbalanced antenna feeding scheme significantly

improves the antenna's performance. Impedance mismatch is mitigated by utilizing the FPRFS to produce matching circuits. Our proposed FPRFS is a solution to overcome traditionally undesired unbalanced feeding scheme issues, enhancing antenna efficiency to the point where it can function as a productive antenna. This solution eliminates the limitations imposed by the number of RF switches on compromised radiation performance, unlocking significantly increased potential in flexibility, programmability, and scalability for future reconfigurable antennas.

In addition to implementing the antenna radiating structure, the FPRFS also has the advantage of implementing the impedance-matching networks (IMN) to maximize the amount of incident power absorbed by the antenna load. Reconfigurable IMNs have received limited attention and have only been reported as accessories with limited reconfigurability for other RF devices[29–32]. Here, the small reconfigurable transmission line segments on the FPRFS can be configured to implement matching stub impedance tuning networks where the location, the number of stubs, and the length of each stub are independent parameters that can be chosen in real time.

In this work, we proposed manipulating the RF performance characteristics of antennas and impedance-matching networks by modifying the current distribution on a conductive surface. We provided a mathematical foundation, and the model was verified experimentally. We also implemented the FPRFS hardware with a pixelated surface structure and demonstrated it as a reconfigurable antenna and impedance-matching network. We hypothesized and experimentally verified improved FPRFS-based antenna efficiency using an asymmetric excitation method. The physical mechanism for improved performance was also investigated.

## Results

### Theoretical formulation for FPRFS antennas

For the FPRFS shown in Fig. 1b, the yellow segments denote all small current elements, and red denotes the segments that have been activated and are part of the surface-carrying current. A solid ground plane is located on the other side, and a dielectric slab is located in between. Each transmission line segment is modeled as a small dipole with a uniform current distribution. The far-field electromagnetic field from a small current dipole is determined, and the resultant fields are determined by integrating over the spatial distribution of current flowing on all the dipole segments.

The far-field radiation characteristics of a line current given its dimension and current density can be easily derived. For a small current dipole with a length $\lambda/50 < l_0 < \lambda/10$, the uniform constant current distribution approximates the actual current distribution[25]. Assuming the dipole length to be $l_0 = 10$ mm, it fulfills the small dipole requirement in the ultra-high-frequency (UHF) range. With an arbitrary given number of segments $n$, the total vector potential function can be expressed as[25]

$$\mathbf{A}(x,y,z) = \sum_{i=1}^{n} \mathbf{A}_i(x,y,z) = \sum_{i=1}^{n} \mu I_i l_0 \cdot e^{-jkr_i}/8\pi r_i \cdot \hat{\mathbf{a}}_i, \tag{1}$$

where $I_i$ is the phasor current magnitude in the dipole segment $i$, $k$ is the wave number, $r_i$ is the distance from the observation point to the dipole segment $i$ center, and $\hat{\mathbf{a}}_i$ is the unit vector along its phasor current direction. The H field and E-field can be derived as follows:

$$\mathbf{H} = \frac{1}{\mu}\nabla \times \mathbf{A}, \mathbf{E} = \frac{1}{j\omega\varepsilon}\nabla \times \mathbf{H}. \tag{2}$$

To simplify the analysis, the ground plane is approximated as an infinite perfect electrical conductor (PEC) located at a distance below the loop antenna. The permittivity of both the substrate and space are assumed to be $\varepsilon_0$. Based on image theory[25], the array factor of the superposition of the real dipole current source and its virtual dipole current source is $2j\sin(kh\cos\theta)$, derived in Supplementary Note 2. The E and H fields are zero for $z < 0$. Thus,

$$\mathbf{H}_{\text{total}} = \begin{cases} 2j\sin(kh\cos\theta)\cdot\mathbf{H} & z \geq 0 \\ 0 & z < 0 \end{cases}, \tag{3}$$

$$\mathbf{E}_{\text{total}} = \begin{cases} 2j\sin(kh\cos\theta)\cdot\mathbf{E} & z \geq 0 \\ 0 & z < 0 \end{cases}. \tag{4}$$

The average Poynting vector is equal to:

$$\mathbf{W}_{\text{av}} = \frac{1}{2}\text{Re}[\mathbf{E}_{\text{total}} \times \mathbf{H}_{\text{total}}^*], \tag{5}$$

and the radiation intensity at $z \geq 0$ is equal to:

$$U = r^2 W_{\text{av}}. \tag{6}$$

Without loss of generality, we will initially consider a loop antenna pattern on the FPRFS with nine activated rectangular transmission line segments and its simplified model for mathematical analysis, as shown in Fig. 2a. The small dipole modeling and calculation process are thoroughly outlined in the Supplementary Note 3.

The calculated radiation patterns in the $\phi = 0°$ and $\phi = 90°$ planes at 2.45 GHz are shown in Fig. 2b with solid curves, and the HFSS simulated radiation patterns are plotted by dotted curves. The process is carried out for two other loop antenna structures with eleven activated segments, as shown in Fig. 2c, d, and the radiation pattern results

are shown in Fig. 2e, f. Their total H field distributions are calculated accordingly in Supplementary Note 3. Consistent results are observed between theoretical calculations and simulations. Based on the analysis's infinite PEC ground plane assumption, there is no radiation in the negative semi-sphere. Radiation pattern variation is observed among different loop antenna patterns in the positive semi-sphere. The minor differences between theoretical and simulation results are caused by the ideal line current distribution assumption and ignored RF switches, dielectric layer, and mutual coupling in the mathematical model. Not surprisingly, the results confirm that choosing rectangular conductors as pixels is a justified choice and demonstrate that the far-field radiation properties of the antennas can be determined by integrating the fields generated by decomposed IDMs.

### Theoretical formulation for FPRFS impedance-matching networks

Impedance matching is essential for reducing the voltage standing wave ratio (VSWR), maximizing the transferred power to the load, and protecting the source[33]. Conventional IMNs are designed for a specific load and have limited bandwidth. However, load impedance may change due to active components in the load[34] or environmental factors[35]. The FPRFS can be programmed to function as IMNs by utilizing the rectangular conductor pixels to implement transmission lines and tuning stubs. To optimize the pixel dimension, an impedance-matching capability evaluation standard for the FPRFS is formulated based on a classic impedance-matching analytical tool, the Smith chart, and tuning theory[36]. A perfect match is achieved when the impedance is at the center of the normalized Smith chart. However, in practice, a perfect match is not always achievable. Thus, a concentric circle with a radius $k$ on the Smith chart is defined as the acceptable matched region. For a given radius $k$, the corresponding VSWR and return loss (RL) can be expressed as:

$$\text{VSWR} = \frac{1+k}{1-k}, \text{RL} = -20\log_{10}k. \tag{7}$$

For the following analysis, we select $k = 0.2$, VSWR = 1.5, and RL = 14 dB. Consider a main transmission line with six segments as shown in Fig. 3a. There are 25 single-stub matching patterns and 250 double-stub matching patterns. The tuning stub lengths are integer multiples of the unit length $l_{\text{unit}}$, consisting of the rectangular pixel length $l_0$ and the square pad width $W$, as shown in Fig. 3a. The effects of the RF switches are ignored for this theoretical analysis. $W$ is designed to achieve the 50 ohm characteristic impedance as shown in Supplementary Note 4.

Consider a specific double-stub tuning pattern as shown in Fig. 3a. The equivalent circuit diagram is shown in Fig. 3b, and the impedances looking from the arrows in Fig. 3b, are plotted with the corresponding colors in Fig. 3c, illustrating the impedance-matching methodology on the Smith chart. As we defined an acceptable matched region on the Smith chart (as the green concentric circle denotes) instead of a perfect match denoted by a single point at the center[29], the whole impedance-matching process is illustrated by concentric circle regions instead of individual points. Assuming the range of load admittance can be matched by a specific double-stub IMN pattern to be an unsolved variable $Y_L$, the admittance at a distance $d_1$ to the load is equal to:

$$Y_{\text{in\_d1}} = Y_0 \frac{Y_L + jY_0\tan\beta d_1}{Y_0 + jY_L\tan\beta d_1}, d_1 = m_1 l_{\text{unit}}, m_1 = 1,2,3,4. \tag{8}$$

The wave number $\beta$ should take into account the effective dielectric constant[36] as derived in Supplementary Note 5. The

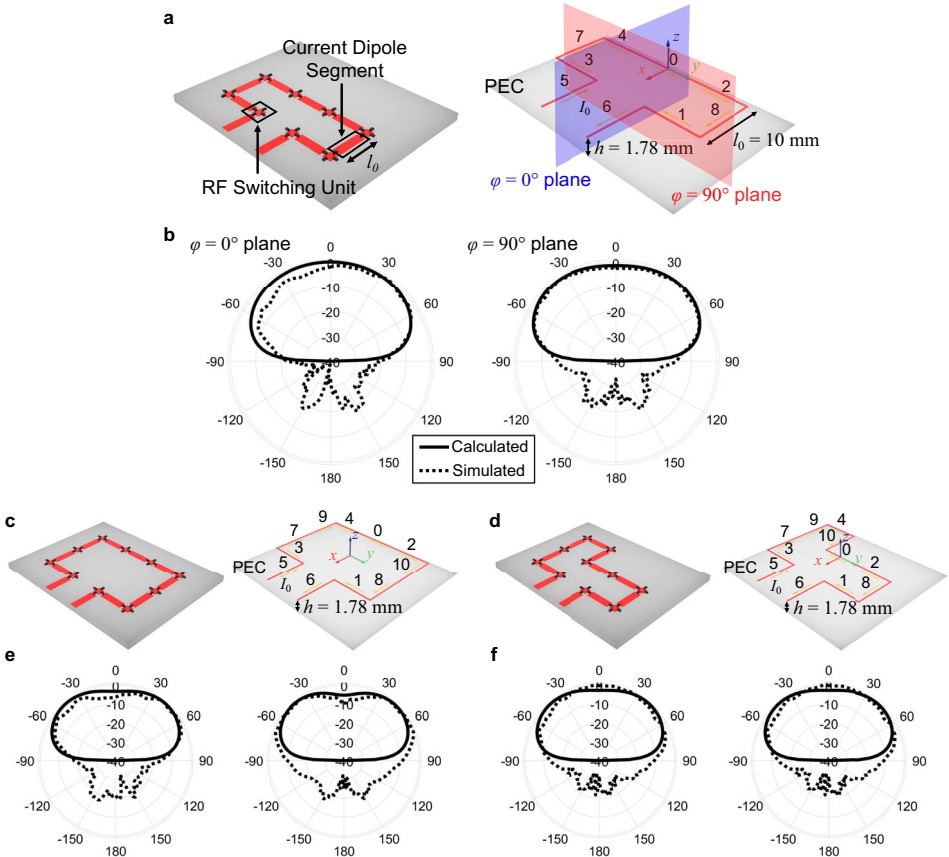

**Fig. 2 | Theoretical analysis for FPRFS antennas. a** The structure of an FPRFS emulated loop antenna and its corresponding ideal line-current model over a perfect electronic conductor (PEC). **b** The calculated and simulated radiation patterns for the loop antenna models at 2.45 GHz. **c, d** The structures of two other FPRFS emulated loop antennas and their corresponding ideal line-current models. **e, f** The calculated and simulated radiation patterns for the other two loop antenna models at 2.45 GHz.

admittance of the first open stub to the load with a length $l_1$ is equal to:

$$Y_{\text{in}\_l1} = jY_0 \tan\beta l_1, l_1 = n_1 l_{\text{unit}}, n_1 = 1,2,3,4,5. \quad (9)$$

Thus,

$$Y_{\text{total1}} = Y_{\text{in}\_d1} + Y_{\text{in}\_l1}. \quad (10)$$

Regarding $Y_{\text{total1}}$ as the new load admittance and repeating the process:

$$Y_{\text{in}\_d2} = Y_0 \frac{Y_{\text{total1}} + jY_0 \tan\beta d_2}{Y_0 + jY_{\text{total1}} \tan\beta d_2}, d_2 = m_2 l_{\text{unit}}, m_2 = 1, \ldots, 5 - m_1, \quad (11)$$

$$Y_{\text{in}\_l2} = jY_0 \tan\beta l_2, l_2 = n_2 l_{\text{unit}}, n_2 = 1,2,3,4,5, \quad (12)$$

$$Y_{\text{total}} = Y_{\text{in}\_d2} + Y_{\text{in}\_l2}. \quad (13)$$

Solving the inequality below for $Y_L$ and one circle of impedances on the Smith chart that can be matched yields:

$$|\Gamma| = \left| \frac{Y_0 - Y_{\text{total}}}{Y_0 + Y_{\text{total}}} \right| \leq k. \quad (14)$$

Each FPRFS IMN pattern corresponds to a $Y_L$ circle. After solving all 275 IMN pattern corresponded inequalities, the total matchable load impedance area was derived as described in Supplementary Note 6 and outlined by the red contour as shown in Fig. 3d. The FPRFS's

impedance-matching capability is defined as the percentage of this area to the total Smith chart area. The impedance-matching capability is a function of the unit length $l_{\text{unit}}$ and frequency $f$. The selection range for pixel length is not further reduced to prevent the dominance of PIN diode-based RF switches in determining the electrical length (Supplementary Note 1). At 2.45 GHz, the relationship between $l_{\text{unit}}$ and the impedance-matching capability is plotted in Fig. 3e. To ensure that the small dipole condition discussed in FPRFS antenna analysis is met, $l_{\text{unit}} = 14$ mm and $l_0 = 10$ mm are finally chosen, achieving the highest impedance-matching capability of 73.55%. The impedance-matching capability as a function of frequency is plotted in Fig. 3f. The results indicate that the FPRFS has an impedance-matching capability of over 50% in approximately 80% of the analyzed spectrum. The impedance-matching bandwidth is an important metric to assess the reconfigurable IMN's performance. The dynamic and static impedance-matching bandwidths are defined in Supplementary Note 7, with accompanying examples provided. Theoretical analysis determines the optimal pixel shape and dimensions of the proposed FPRFS.

**Operating frequency characteristics of FPRFS patch antennas**
The FPRFS planar structure, as shown in Fig. 4a, consists of an array of cruciform RF switching units as the basic blocks. Each unit comprises four RF switches that connect four adjacent transmission line segments with length $l_0 = 10$ mm, to a shared middle square pad, as highlighted by the red square. Various RF switches, including RF-MEMS[37,38], PIN diodes[39,40], and liquid metal[41,42] are generally used for reconfigurable RF devices[43]. A PIN diode is used in this work due to its low cost, fast switching speed, simple simulation model[44,45], and decent performance[46]. Vertically, a solid internal ground plane is

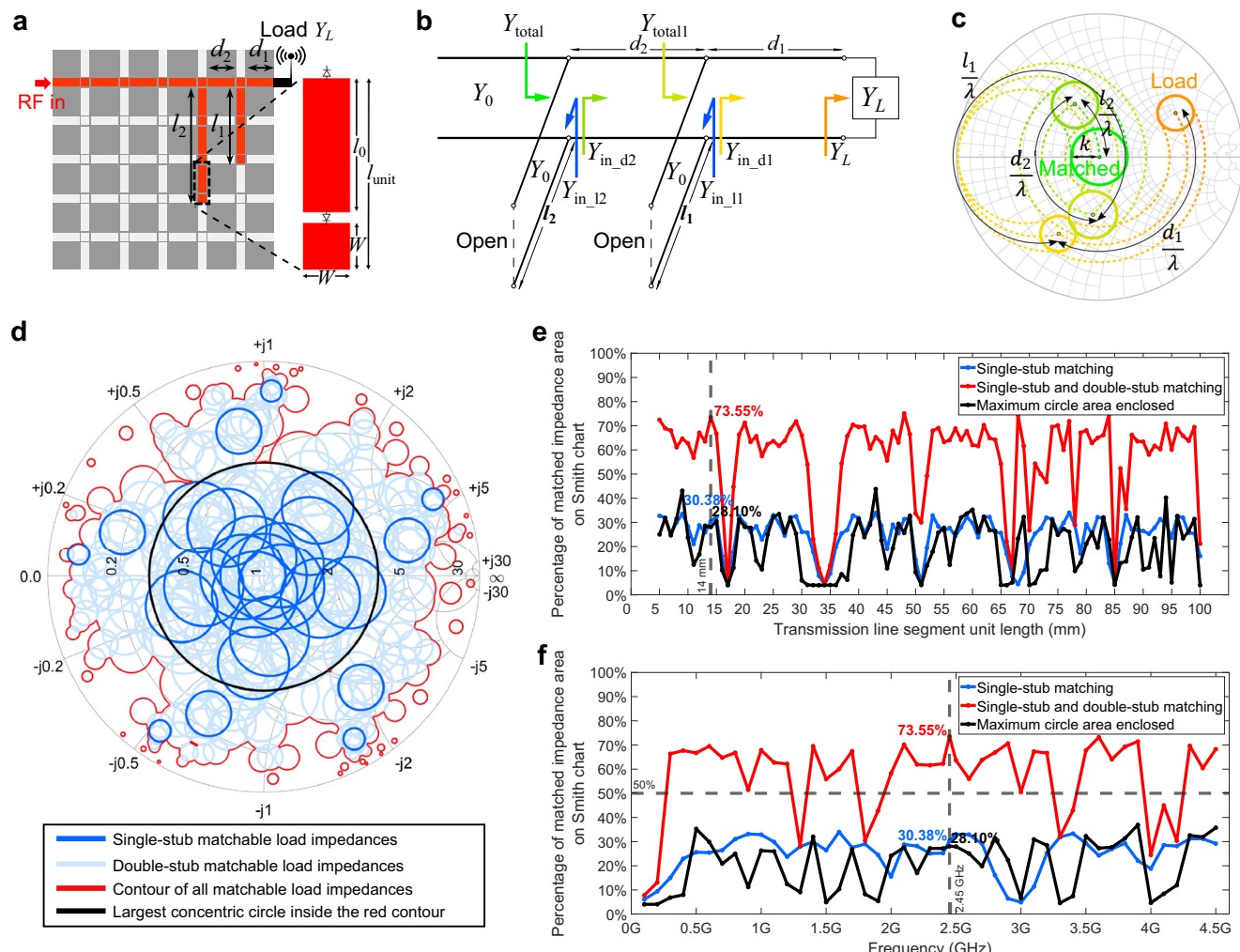

**Fig. 3 | Impedance-matching capability analysis for FPRFS IMNs. a** A double-stub IMN pattern programmed on the FPRFS. **b** The circuit model of the FPRFS emulated double-stub IMN. **c** The impedance-matching process illustration of the double-stub IMN on the Smith chart. **d** All matchable load impedance circles plotted on the Smith chart. **e** The function of impedance-matching capability versus the segment unit length. **f** The function of impedance-matching capability versus the operating frequency.

located 1.78 mm beneath the RF surface layer, whilst the biasing circuits with onboard memory are located on the bottom side, as shown in the inset of Fig. 4a. The onboard memory holds the biasing states of all PIN diodes, which enables the RF conductive surface pattern to be digitally programmed using a binary bitstream.

As the RF switches segment the top RF surface, the inflow RF current experiences energy losses[47], reducing far-field radiation gain and efficiency. As discussed in the next section, since any RF current has an imaged current flow on the ground plane, we intentionally inverted the signal and ground polarity for improved radiation performances. The FPRFS input and output (IO) ports are fed using coaxial cable and SMA connectors. The feeders with inverted polarity are shown in Fig. 4b. As the inverted polarity FPRFS antenna schematic shows in Fig. 4c, the signal flows into the solid low-loss plane, and its current distribution is manipulatable as it closely resembles an image of the current flowing on the segmented programmable RF surface as the ground. However, the ground plane becomes lossy[48] and high-impedance[49], which is ineffective at acting as a sink for current flow[50]. This creates an unbalanced antenna structure with increased return current on the coaxial outer side[51], which is commonly avoided by using baluns[52]. Problems, including unexpected return current radiation and susceptible VSWR, arise[53]. However, these issues can be addressed by the FPRFS's self-adaptive capability, as discussed in the following sections.

The FPRFS's capability to emulate a conventional rectangular patch antenna was explored by measuring the reflection coefficients of 80 possible patch antenna patterns with varying lengths, widths, feeding positions, and feeding line lengths. These parameters uniquely named each pattern. An example of a 1 × 2-2-1 (patch length equals 1-segment; patch width equals 2-segment; feeding position is the 2nd from the patch corner with a feeding line length of 1-segment) patch antenna pattern is shown in Fig. 4a. The results for FPRFS emulated patch antennas with inverted polarity are shown in Fig. 4d, a red dashed curve enveloping the S11 valleys shows a wide dynamic bandwidth of 3 GHz. At least one patch antenna pattern can be programmed to work at any target operating frequency inside the dynamic bandwidth. To demonstrate this capability, a typical 1 × 2-2-1 patch antenna with both non-inverted polarity (NIP) and inverted polarity (IP) was simulated and experimentally measured, along with the conventional patch antenna reference (REF). The results in Fig. 4e show similar S11 responses, with all having three resonant frequencies, although with slight frequency shifts. Figure 4f–h show the patch cavity field distributions used to gain further insights into antenna operating modes[54]. The distributions revealed highly consistent electric field patterns among IP, NIP, and REF patch cases at three S11 valleys, indicating identical patch antenna operation modes of TM$_{10}$, TM$_{02}$, and TM$_{12}$. The FPRFS emulated patch

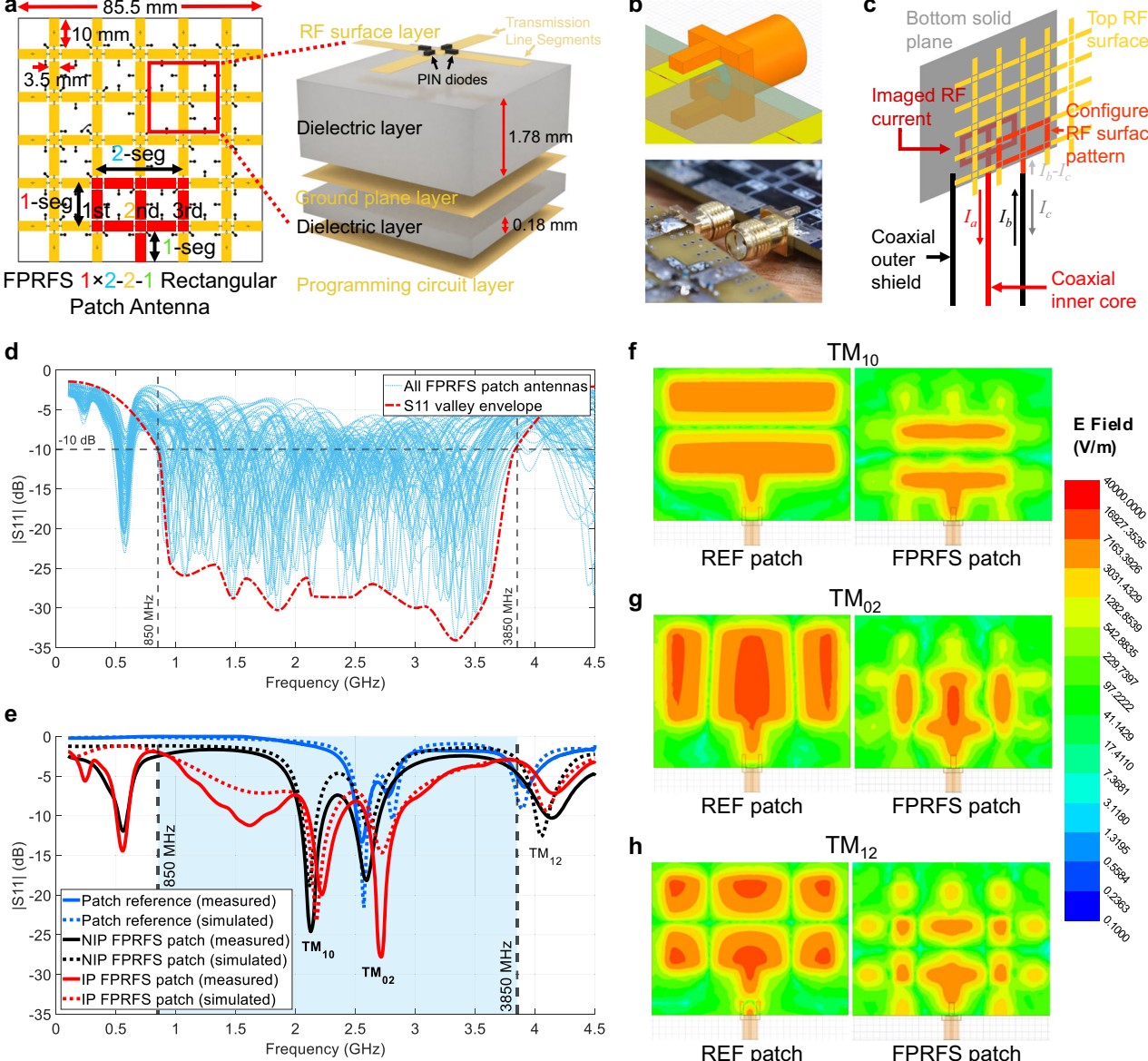

**Fig. 4 | FPRFS structure and FPRFS emulated patch antennas. a** The planar structure of the pixelated RF surface with a programmed 1 × 2-2-1 patch antenna and the vertical structure of the FPRFS. **b** A mounted SMA connector on the FPRFS with inverted polarity. **c** The schematic of an FPRFS-based antenna with inverted coaxial feeder polarity (IP). **d** Reflection coefficient frequency span of all FPRFS emulated IP rectangular patch antennas. **e** Reflection coefficients of the FPRFS emulated and conventional 1 × 2-2-1 patch antennas. **f–h** The electric field cavity mode comparisons between the conventional reference patch and the FPRFS emulated patch antennas at three transverse magnetic (TM) modes.

antennas have a smaller cavity size than their REF patch counterparts because the PIN diodes conducting current inside the active pattern have a larger equivalent electrical length than their physical lengths. For patch antennas, the energy is radiated from the edge slot[25]. Despite the square hollows in the FPRFS emulated patch, the antenna shape outlined by the rectangular segments still predominantly radiates from the edges, thus maintaining the operating modes. The high consistency between all simulated and measured S11 curves supports the adequacy of our proposed models for the FPRFS. These results demonstrate the FPRFS's operating frequency programmability with wide dynamic bandwidth and its capability of emulating conventional patch antennas with arbitrary shapes. Its radiation pattern programmability is demonstrated in the following sections and the polarization programmability is discussed in Supplementary Note 8.

## Improved reconfigurable antenna performance with inverted polarity

The FPRFS emulated patch antennas with IP exhibit substantial improvements in radiation gain, efficiency, and immunity to obstacles. The radiation patterns of the FPRFS emulated 1 × 2-2-1 patch in H-plane and E-plane were measured at these three operating modes for both NIP and IP and compared with the simulated results as shown in Fig. 5a–c. At the dominant TM$_{10}$ mode, the energy radiated with the NIP feeder mainly concentrates in the upper semi-sphere with a maximum gain of −10 dBi. While for the IP counterpart, the largest measured gain of around 0 dBi is observed directly below the FPRFS. This gain boost in the lower semi-sphere results from the imaged signal current distribution on the bottom low-loss solid plane. In the TM$_{02}$ mode, as shown in Fig. 5b, where significantly less energy is radiated by the NIP, the IP achieves an average 15 dBi greater measured gain in all

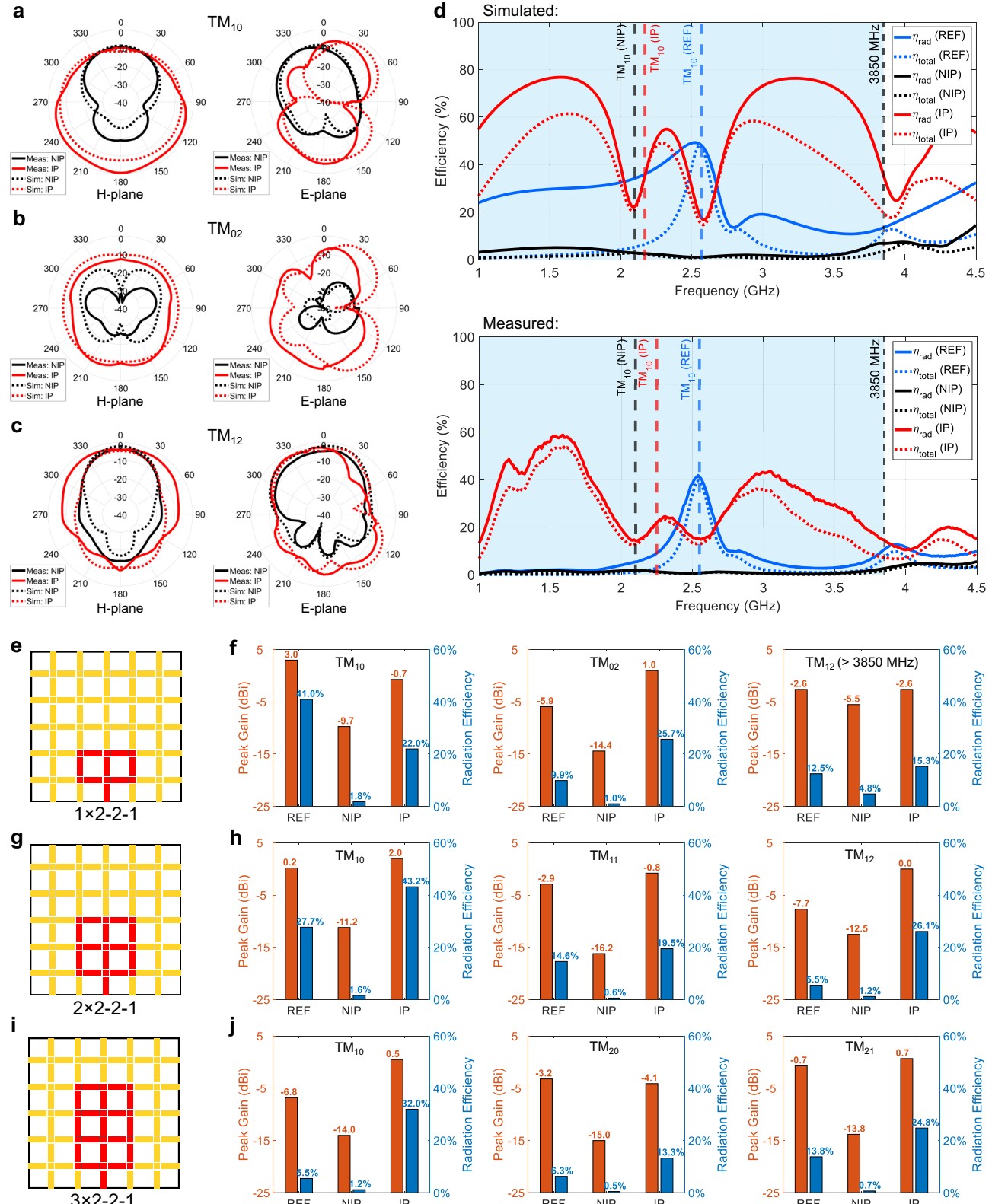

**Fig. 5 | Enhanced radiation performance of FPRFS antennas with the inverted polarity (IP) feeding scheme. a–c** The measured and simulated radiation patterns for both non-inverted polarity (NIP) and IP cases in three transverse magnetic (TM) modes. **d** Simulated and measured radiation and total efficiency results of REF, NIP, IP 1×2-2-1 patch antennas as functions of frequency. **e** The FPRFS 1 × 2-2-1 antenna pattern and **f** the peak gain and radiation efficiency comparisons in three TM modes. **g** The FPRFS 2×2-2-1 antenna pattern and **h** the peak gain and radiation efficiency comparisons in three TM modes. **i** The FPRFS 3 × 2-2-1 antenna pattern and **j** the peak gain and radiation efficiency comparisons in three TM modes.

directions. For the $TM_{12}$ mode, located outside the FPRFS antennas' dynamic bandwidth at around 4.2 GHz, the radiation gains for both NIP and IP are larger, but the gain increase achieved by IP is less due to poor PIN diode isolation at high frequencies, as shown in Supplementary Fig. 11.

Figure 5d illustrates a high consistency between simulated and measured efficiency results. With the IP feeding scheme, both the radiation and total efficiencies surpass those with NIP feeding by more than tenfold, consistently maintaining elevated levels (above 30%) across a wide dynamic bandwidth. The IP also outperforms conventional patch references, which achieve high total efficiency solely at their principal operating frequency. Figure 5e–j show the measured peak gains and radiation efficiencies of three patch antenna patterns with different patch lengths and numbers of lossy RF switches, addressing the differences among the REF, NIP, and IP at three operating modes. As the patch length increases, the dominant $TM_{10}$ mode frequency decreases and the number of lossy RF switches increases. As a result, the radiation efficiency of the FPRFS patch with NIP decreases to an even lower level of about 1%. However, for the FPRFS patch with IP, the radiation efficiency remains mostly above 30% without significant degradation in $TM_{10}$ mode. Moreover, except for the mode located at high-frequency, where the PIN diode OFF behavior is poor, both the peak gain and radiation efficiency achieved with the IP FPRFS are much higher than those achieved with the NIP FPRFS, and they are comparable to those of their conventional patch antenna counterparts. The antenna peak gain and efficiency comparison plots in relation to frequency can be found in Supplementary Fig. 16. This insight implies that the FPRFS with the inverted polarity approach can emulate patch antennas of arbitrary geometries by incorporating many more pixels and RF switches while maintaining uncompromised antenna performances. A table comparing our work to others is included in Supplementary Note 12.

Another advantage is the high immunity to environmental obstacles of FPRFS emulated patch antennas with IP. Metal plates that block a part of and the whole FPRFS proximate areas are attached to emulate obstacles. The obstacles are applied one at a time to both the top and bottom sides of the FPRFS, as shown below. The $1 \times 2\text{-}2\text{-}1$ rectangular patch antenna is programmed on the FPRFS, and the measured $TM_{10}$ mode radiation patterns without obstacles are plotted by the black dotted curves in Fig. 6b, d. When the top side of the NIP antenna is blocked by an obstacle, the antenna gain in most of the upper semi-sphere suffers a significant degradation of greater than 10 dBi, as shown in Fig. 6b, d. In contrast, for the IP antenna, no substantial gain degradation is observed. This illustrates that the radiation pattern of the NIP antenna is highly vulnerable to obstacles above it, whereas the IP antenna is much less susceptible. It is also important to note that entirely blocking the ground plane of the IP FPRFS results in an average 5 dBi decrease in gain, further verifying that a substantial amount of radiation emanates from the FPRFS's bottom solid plane in IP cases.

This significantly improved radiation performance is analyzed by comparing the electric field (E-field) and surface current (J-surf) distributions on the antenna and the feeder surfaces, as shown in Fig. 6e. In all modes, the magnitudes of E-field and J-surf on the top segmented surface are larger than those on the bottom solid surface, even for IP cases where the bottom surface radiation accounts for a significant proportion. The magnitudes of E-field and J-surf on both surfaces in IP cases are about half of those in NIP cases. Therefore, the increased radiation gain and efficiency must result from other sources. Based on previous analysis, the return current originates on the feeder's outer side when the antenna is unbalanced. As seen in the E-field distributions shown in Fig. 6e (column three) and J-surf distributions shown in Fig. 6e (column six), both exhibit an increase of more than an order with IP compared to the NIP counterparts. In the high-frequency $TM_{12}$ mode, these distribution differences between NIP and IP cases are much smaller, which also corresponds to much smaller radiation performance differences, due to the poor OFF behavior of the RF switches. Unlike traditional coaxial feeders that utilize the outer metallic shield for grounding and shielding against electromagnetic radiation, the inverted polarity technique transforms the feeder into an integral part of the antenna radiator by enhancing the return current. The return current distributed along the cylindrical coaxial feeder explains the more omnidirectional H-plane radiation patterns and higher immunity to obstacles covering the antenna surfaces.

For conventional antennas with fixed structures, the return current enhancement triggered by inverting polarity should be avoided, as its radiation interferes with the delicately designed radiation characteristics of the primary antenna. To use the additional feeder radiation, it is essential to ensure that the feeder radiation does not dominate the total radiation. This ensures the manipulability of the total radiation through programming the FPRFS. Comparing the J-surf magnitudes on the IP FPRFS antenna surfaces and the coaxial cable surface, the magnitudes of the latter are still about an order of magnitude smaller than the former. This implies that the manipulable FPRFS J-surf distribution still significantly determines the antennas' far-field radiation characteristics. Further insights will be revealed in the next section through experiments.

## FPRFS self-optimization and self-adaptiveness based on environmental conditions

Although traditionally, the return current on the feeder has been demonstrated to improve the radiation performance of FPRFS-based reconfigurable antennas, challenges related to unexpected radiation patterns and shifts in VSWR arising from the return current need to be addressed. As theoretically analyzed before, the conceptual FPRFS can be modified to emulate a desired current distribution to achieve the desired radiation pattern, while also providing good impedance-matching capabilities to cover a wide range of VSWR cases. Practically, an FPRFS control and visualization software interface has been developed, running surface pattern-sweeping algorithms to find the optimal FPRFS pattern. In this work, a brute force algorithm automatically sweeps a range of FPRFS patterns quickly, as demonstrated in Supplementary Movie 1 for antenna sweeping and Supplementary Movie 2 for IMN sweeping. The potential of applying advanced optimization algorithms, such as Artificial Neural Networks (ANNs), is discussed in Supplementary Note 13. Feedback sensing and control are essential components of adaptive wireless communication systems[55], such as beamforming antennas[56] and cognitive radio systems[57], facilitating optimal decision-making processes. They are integrated with our FPRFS for self-adaptive applications. By sensing different RF characteristic parameters as feedback, the software finds the optimal FPRFS pattern under the prevailing environment based on these criteria.

For radiation pattern optimization, a key parameter is the gain. Hence, the feedback sensing unit comprises a gain sensing antenna and an RF meter, as illustrated in the experimental setup in Fig. 7a. Depending on the relative position between the FPRFS and the sensing antenna, the antenna gain in that direction is optimized. Using more sensing antennas in other directions helps with shaping the overall radiation pattern comprehensively. In the experiment, the FPRFS antenna is rotated in the H-plane to simulate different relative positions, and the sweeping-based self-optimization process is carried out at 2.45 GHz for each rotation angle to find the optimal FPRFS antenna pattern, as depicted in Fig. 7b. The measured radiation patterns for these algorithmically determined FPRFS antenna patterns are shown in Fig. 7c and compared with the radiation pattern of the FPRFS in an idle state where no active conductor pattern is configured and only the feeder return current exists. The result indicates that despite the presence of the return current with IP, its radiation does not dominate, and the radiation pattern of the FPRFS antenna remains manipulable.

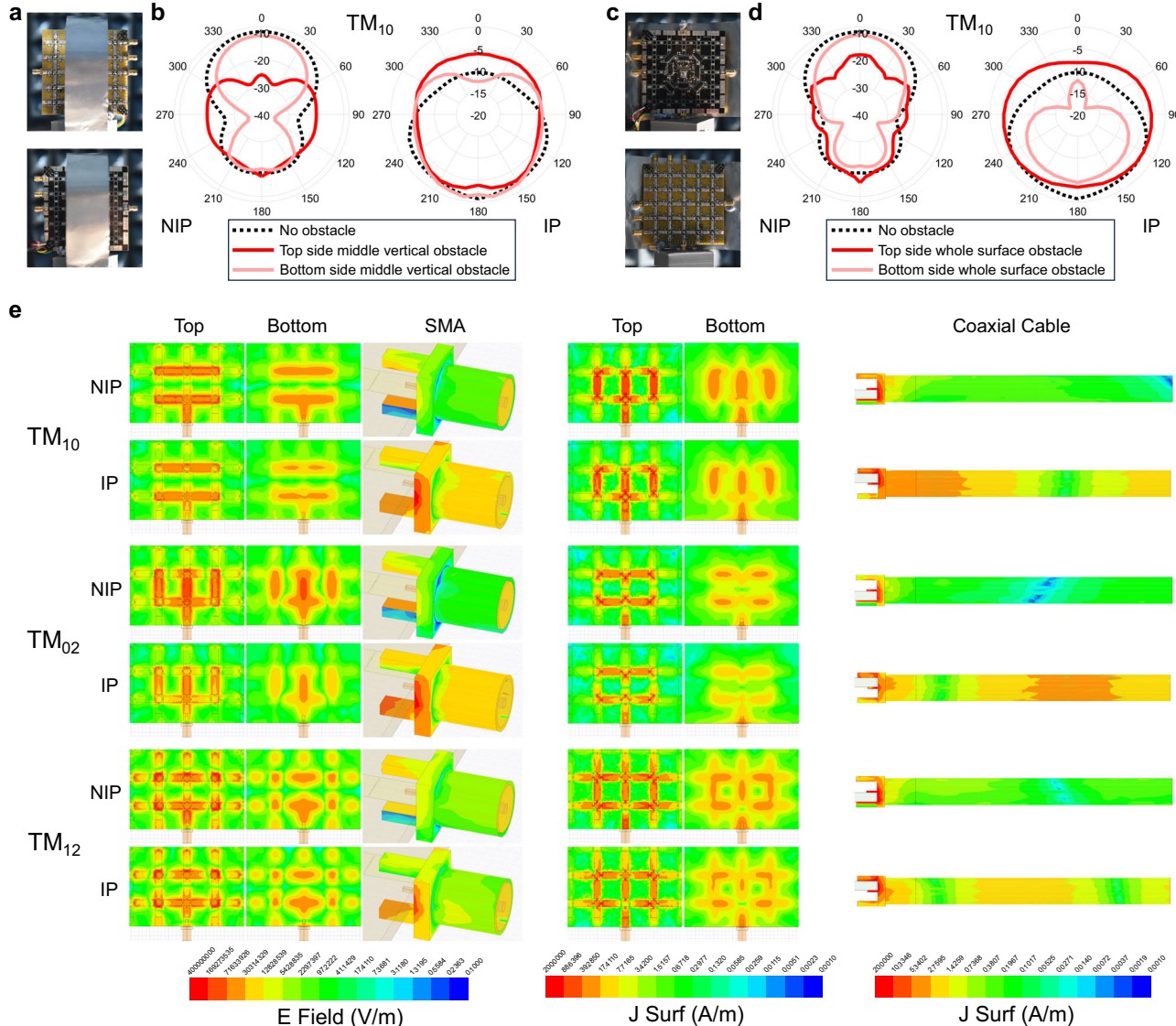

**Fig. 6 | Discussion on enhanced radiation performances. a** The 1 × 2-2-1 FPRFS antenna with partial proximity surface blocking on the top side or bottom side. **b** The measured radiation patterns at TM$_{10}$ mode with non-inverted polarity (NIP) feeding and inverted polarity (IP) feeding. **c** The 1 × 2-2-1 FPRFS antenna with whole proximity surface blocking on the top side or bottom side. **d** the measured radiation patterns at TM$_{10}$ mode with NIP feeding and IP feeding. **e** Visualized electric field (E-field) and surface current (J-surf) distributions on the FPRFS and the feeders in three transverse magnetic (TM) modes for the FPRFS emulated 1 × 2-2-1 antenna.

Hence, the unexpected radiation pattern issue becomes negligible with the introduction of the FPRFS featuring radiation pattern reconfigurability and a self-optimization strategy.

For addressing susceptible VSWR, a critical parameter is the reflected power. In this experiment, an FPRFS is programmed to implement an IMN to adjust and minimize the VSWR. As shown in Fig. 7d, two FPRFSs are cascaded, with the highlighted FPRFS #1 as an IMN running the sweeping algorithm and FPRFS #2 functioning as the 1 × 2-2-1 patch with IP. For testing convenience, a vector network analyzer (VNA) is used to both source the signal to the system and analyze the reflected signal power coupled from a directional coupler. To eliminate the systematic error brought by the directional coupler, it is modeled with insertion loss IL($f$), isolation IS($f$), and coupling CP($f$) as shown in Fig. 7e. Its S-parameter characterization is shown in Supplementary Note 14. The calibrated load response is thus expressed by:

$$H(f) = \frac{V_2(f)/V_1(f) - \mathrm{IS}(f)}{\mathrm{IL}(f) \cdot \mathrm{CP}(f)} = \frac{S_{21}(f) - \mathrm{IS}(f)}{\mathrm{IL}(f) \cdot \mathrm{CP}(f)}. \tag{15}$$

Setting three target frequencies at 1.2 GHz, 1.8 GHz, and 2.4 GHz, respectively for VSWR tuning, the optimized S11 are all below −14 dB, as shown in Fig. 7f, achieved by the corresponding algorithm-found IMN patterns as shown in Fig. 7g. The corresponding VSWRs are equal to 1.12, 1.11, and 1.18, all located inside the previously defined acceptable matched region. The single-stub impedance-matching process at a target frequency of 2.4 GHz is shown in Supplementary Movie 3. The results demonstrate the FPRFS's remarkable capability to match impedance and reduce VSWR. In practical scenarios with sources featuring complex frequency components and requiring a fast response, an RF meter-based setup, illustrated below, is compact, has fast response speed, and offers versatility. Their differences are further discussed in Supplementary Note 15. The RF meter-based setup process for the same single-stub matching test, at 2.4 GHz, is shown in Supplementary Movie 4 with a much faster response. To ensure that the optimal configuration considers multiple parameters, like both SNR and VSWR, multi-objective optimization (MOO) methods can be integrated and are discussed in Supplementary Note 16.

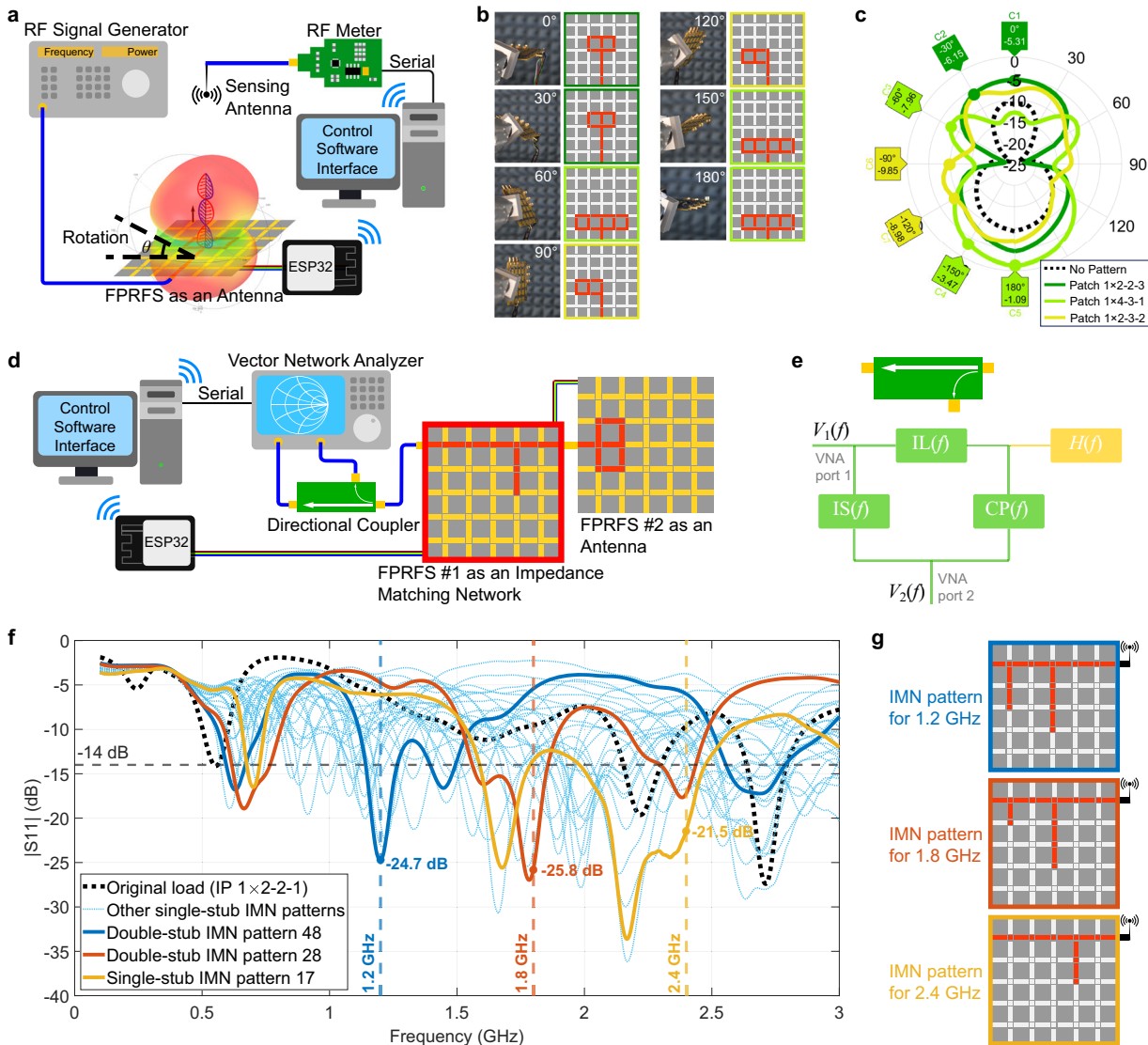

**Fig. 7 | Self-optimization for radiation pattern and voltage standing wave ratio (VSWR). a** The experimental setup for optimizing radiation patterns in different directivities. **b** The algorithm-found FPRFS antenna patterns at different relative angles between the FPRFS and the sensing antenna in the H-plane at 2.45 GHz. **c** Measured radiation patterns of all algorithm-found FPRFS antenna patterns at 2.45 GHz. **d** The experimental setup for optimizing the VSWR of an FPRFS antenna cascading a second FPRFS as an impedance-matching network (IMN). **e** The transfer function model of the directional coupler. **f** The optimized S11 curves at 1.2 GHz, 1.8 GHz, and 2.4 GHz. **g** The corresponding FPRFS IMN patterns.

The FPRFS has demonstrated its capability to effectively mitigate undesirable effects arising from the IP common-mode return current. To adapt to dynamic environments, we introduce a self-adjusting mechanism based on feedback control. Drawing inspiration from standards that implement MIMO antenna beamforming, our FPRFS adjusts the current distribution on the programmable conductive surface, rather than modifying the magnitudes and phases of individual antenna elements. The goal is similar to beamforming antenna systems where the radiation pattern is modified to maximize transmitted power at the receiver. The comprehensive control process, illustrated in Fig. 8a, b, involves channel estimation using Channel State Information (CSI), quantization and encoding of the CSI, feedback transmission, and processing for decision-making. This well-established process has been widely applied in beamforming antennas in IEEE standard[58]. Practically, base stations featuring a larger network coverage and greater processing power function work as the receivers. They can process the CSI comprehensively and send back the optimal pattern index for each mobile station. This strategy holds several advantages. Firstly,

the optimization result takes interference factors into account, focusing on received energy only at target locations of interest. Secondly, it minimizes the spectrum resources needed for transmitting feedback. A distinguishing feature of our work is the introduction of an input impedance-based sensing and adapting feature, where reflected signal strength is used to optimize impedance-matching and minimize VSWR characteristics. The feedback-based self-adaptation feature of the fabricated FPRFS is demonstrated with the setup shown in Fig. 8c. There, the relative station position, interferences, near-field loading effects, and environmental factors such as water on the antenna are all considered, demonstrating the system's ability to adapt to these conditions. Schematic diagrams are presented in Fig. 8d, accompanied by corresponding videos appended as Supplementary Movie 5–10.

The self-optimization capability of the IP antenna to overcome issues and achieve acceptable performance with unbalanced excitation is demonstrated, as is the adaptive capability of the FPRFS to sense interference and rapidly restore the optimal performance in real time.

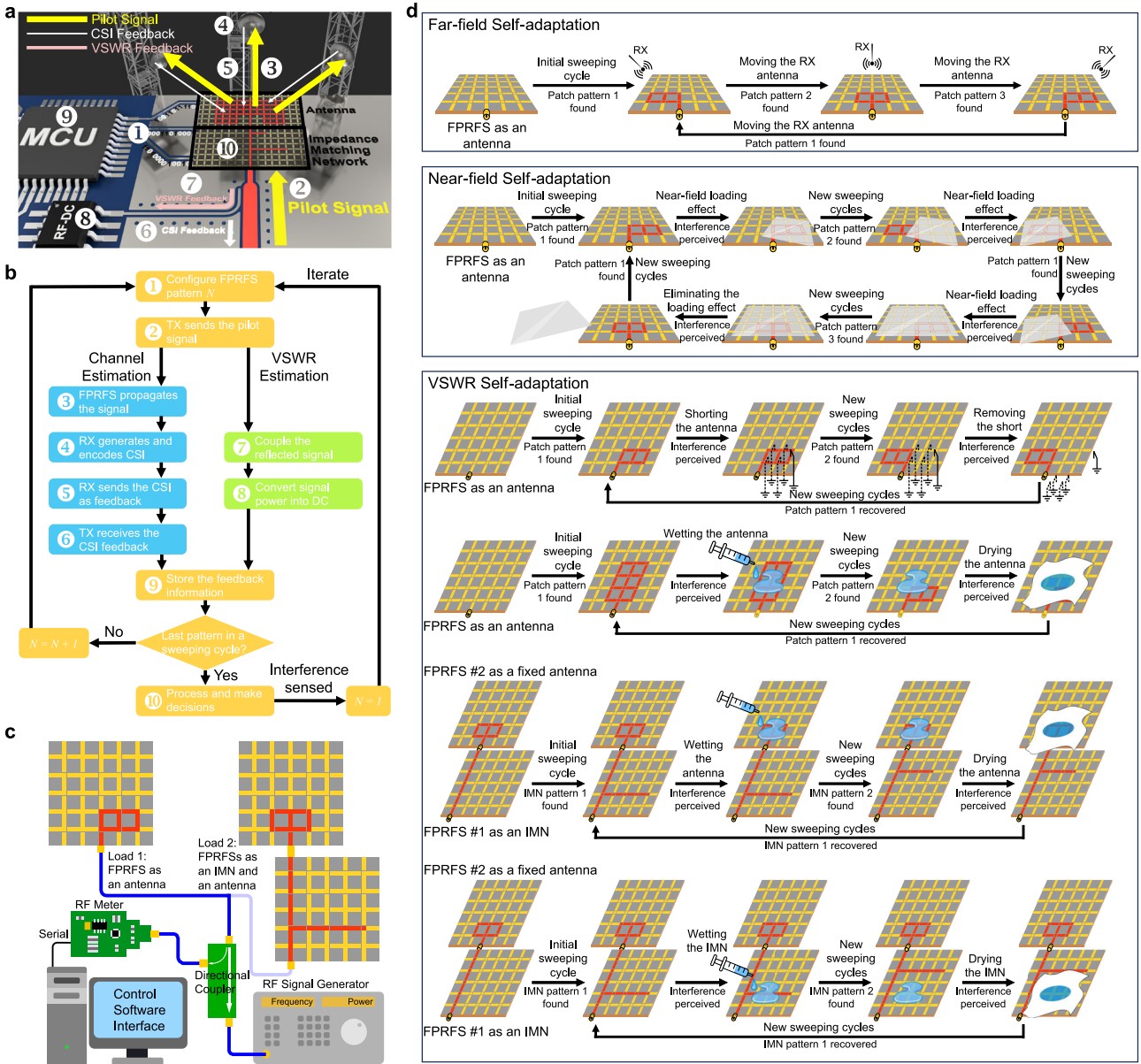

**Fig. 8 | Feedback-based self-adaptiveness. a** The schematic diagram of the feedback control-based self-optimization process. **b** The operation flowchart depicting the operation of the feedback control self-optimization process. **c** The experimental setup for FPRFS self-adaptiveness tests based on near-field and voltage standing wave ratio (VSWR) properties. **d** Experimental illustration diagrams of FPRFS self-adaptation in far-field beam, addressing perturbations in near-field loading effects, shorting circuits damages, and water presence on FPRFS antenna or impedance-matching network (IMN).

## Discussion

This work introduces a Field-Programmable arbitrary RF Surface (FPRFS) and explores its applications in implementing conventional microstrip antennas and impedance-matching networks. The basic programmable element shape and dimensions are determined by mathematical analysis based on infinitesimal dipole modeling for antennas and the impedance-matching capability for impedance-matching networks. It is verified that the far-field characteristics of the FPRFS antennas can be set by manipulating the current distribution on the surface. The conceptual FPRFS is capable of implementing arbitrary conventional patch antennas. Three-dimensional electromagnetic simulations show that the FPRFS-implemented antenna exhibits the same operating modes as a conventional patch antenna. A 3 GHz wide dynamic bandwidth is experimentally demonstrated for the implemented reconfigurable antennas. An important finding of this work is that inverted polarity can overcome the significant losses

caused by the RF switches. When fed with inverted signal-ground polarity, the FPRFS antennas exhibited a significant average gain increase of 10 dBi and more than a 10-fold improvement in efficiency, relative to traditional excitation where the conductors with switches are excited. For the inverted FPRFS, the efficiency remained above 30%, and comparable to conventional patch antennas. Additionally, it demonstrated enhanced immunity to obstacles compared to the symmetric-fed FPRFS. Most importantly, the flexibility is not compromised by the number of lossy RF switches employed, breaking the structure resolution limitation for reconfigurable antennas. Using unbalanced excitation is not without challenges, including unexpected return current, radiation, and variable VSWR. Fortunately, these two challenges are addressed with the self-optimization capability based upon the far-field gain and port-reflected power. Secondly, the FPRFS IMN has an impedance-matching capability to cover a wide range of VSWR variations. Finally, the FPRFS demonstrates intelligence in

perceiving environmental interferences and quickly self-adapting to them with optimized configurations in real time. The enhanced performance and demonstrated self-optimization smartness make the high-resolution FPRFS antenna a viable and productive reconfigurable antenna option.

# Methods

## Mathematical analysis

The series expansion and current dipole image theory calculations are detailed in Supplementary Note 2. The calculation of the far-field radiation patterns of the modeled loop antennas and the solution of inequalities for evaluating the impedance-matching capability was carried out using Wolfram Mathematica. The solutions are transferred into MATLAB for plotting. The Smith chart utility in ADS 2020 is used as an auxiliary tool for IMN analysis.

## Prototype design

The FPRFS consists of three layers of metal, acting as the top programmable RF surface with pixelated conductors, the internal solid plane, and the bottom digital biasing circuit plane, as shown in Fig. 4a. The programmable RF surface is composed of cruciform structures as shown in the red square in Fig. 4a. In each cruciform structure, four adjacent rectangular conductor segments (pixels), whose dimension is justified in the main text, are connected to a shared mid-square conductor using PIN diodes (SMP1345 from Skyworks Solutions Inc.) with a common cathode topology. As shown in Supplementary Note 10, the mid-square conductor is connected to the internal DC ground plane through an RF choke (75 nH, 320 mA, 1.224 ohm, self-resonant frequency of 2.4 GHz) and the cathodes of all PIN diodes are connected to the DC ground. The RF choke blocks the high-frequency RF signal and couples the DC voltage. The four rectangular conductor segments are connected to separate DC biasing sources through RF chokes. When a rectangular conductor segment is DC-biased to 0 V, the PIN diodes at both sides are in the OFF state and the segment is inactivated without inflow or outflow or RF current. When biased to VCC, the PIN diodes at both sides are in the ON state and the current path consisting of the rectangular segment and two mid squares at both ends is activated. The combination of different segment activation states achieves abundant RF surface current distributions. On each IO port segment, a 1 uF capacitor is introduced to block the DC voltage and couple the RF signal. The fabricated and modeled FPRFS is shown in Supplementary Fig. 1.

To achieve full digital programmability for PIN diodes relying on DC biasing, shift registers are used as the onboard memory. On an FPRFS device with 60 rectangular segment conductors, eight 8-bit shift registers (74LVC595AD from Nexperia USA Inc.) are cascaded in series and each bit outputs a DC voltage to bias one segment with two PIN diodes. The digital IC chips are grounded through an RLC circuit as a low-pass filter to isolate the RF noises. Each FPRFS is designed with an input bus and an output bus for multi-FPRFS cascading extendibility. The ground pin of the bus is connected to the internal solid ground plane through a 27 nH, 2 A, 70 mohm RF choke with a self-resonant frequency of 2.75 GHz to block the noisy RF signals. The programming bitstream is sent to the onboard memories by a local ESP32 microcontroller.

For FPRFS antennas with inverted signal-ground polarity, the AC coupling capacitor at the IO port blocks the DC 0 V so the PIN diode connected to it can be properly biased. Rather than being called the ground plane, the internal solid plane is called the signal plane. As the RF AC signal used either from a VNA or an RF source meter in this work has no DC offset, the solid plane in this case still has a DC potential of 0 V, which guarantees the normal operations of the PIN diodes and the onboard digital circuits.

## Prototype simulation and characterization

HFSS 2019 is used for all simulations in this paper and the Agilent E5071C vector network analyzer is utilized for all S-parameter characterizations. The simulation models of the PIN diode in ON/OFF states are shown in Supplementary Fig. 10. The lumped component model values are determined from its datasheet (L = 0.7 nH, Rs = 1.5 ohm, Ct = 1.5 pF, Rp = 5 kohm). The simulated results are compared with the measured results of PIN diode RF switches fabricated on a test board as shown in Supplementary Fig. 11. The ON/OFF behaviors are expressed by S21. Practically, the biasing current affects the PIN diodes' ON/OFF behaviors. The simulated results agree closely with the practical measurements at a biasing voltage of 0.88 V and a biasing current of around 5 mA per PIN diode. Excellent ON/OFF performance with high consistency is observed at frequencies between 600 MHz and 3.8 GHz. A single cruciform RF switching unit with four adjacent segments and four PIN diodes is simulated and measured with consistent results shown in Supplementary Fig. 13. The FPRFS is modeled without the bottom plane with biasing circuits in HFSS. The SMA connectors mounting to the FPRFS in different ways, either with NIP or IP, are modeled. All conductors are assigned with a finite conductivity of 58 MS·m$^{-1}$ and the FPRFS substrate is modeled using FR-4 epoxy material in HFSS SysLibrary. The coaxial cable and SMA connector dielectrics are modeled using Teflon material. Waveport is used to excite the system.

## Prototype fabrication

A commercially available 4-layer FR-4 substrate with a relative dielectric constant $\varepsilon_r = 4.28$, a loss tangent $\tan \delta = 0.02$, and a total thickness of 2 mm is used to fabricate the FPRFS. Only three layers are designed with copper as shown in Fig. 4a. The exposed metals at the FPRFS board edge on the bottom surface are connected to the internal solid ground plane using vias for mounting RF IO connectors. After fabrication, all PIN diodes, RF chokes, and digital components were added by soldering.

## Prototype measurements

Far-field measurements are conducted inside a microwave anechoic chamber. The FPRFS is mounted on a Meca500 robot arm and is connected to Port 1 of the vector network analyzer. A calibrated linear polarized receiving (RX) antenna fixed in the anechoic chamber is connected to Port 2 of the VNA. The far-field gains are measured in the Fraunhofer region of the device under test as derived as follows:

$$d_{\text{reactive}} < 0.62 \sqrt{\frac{D^3}{\lambda}} < d_{\text{Fresnel}} < \frac{2D^2}{\lambda} < d_{\text{Fraunhofer}}. \tag{16}$$

The FPRFS antenna radiation efficiency and peak gain are measured using the MVG StarLab system.

For reflected power-based self-optimization and self-adapting tests as shown in Figs. 7d and 8c, the introduced directional coupler is characterized as shown in Supplementary Fig. 17. For the transferred power-based self-adapting test as shown in Fig. 7a, a monochromatic sinusoid wave outputting from a ROHDE & SCHWARZ RF signal generator is connected to the FPRFS antenna. The Rx sensing antenna is connected to an RF meter from MikroElektronika with a valid frequency range from 1 MHz to 8 GHz. The RF energy received at the sensing antenna end is converted into a DC voltage and is sent back to the software interface as feedback. When using the RF meter together with the directional coupler, the uncalibrated systematic error of the directional coupler may introduce minor experimental variability.

The real-time self-adapting feature is implemented by introducing a feedback control loop, where the sensitivity (error threshold) can be adjusted in the software interface and the feedback sensing unit constantly works to compare the error with the threshold. Once the environmental interference causes a sensed error greater than the threshold, a new cycle of self-optimization will be carried out and the FPRFS pattern countering the interference the best is recovered.

**Prototype programming scheme**

A control and visualization software interface, as illustrated in Supplementary Note 16, was developed based on a MATLAB graphical user interface (GUI) to either manually configure the FPRFS or run a pattern-sweeping algorithm for automatic configuration. Every bitstream consists of initially a byte denoting the number of boards under configuration and the following bytes for determining the FPRFS pattern on each board. Each board introduces a 64-bit bitstream following the initial byte. This bitstream is then transmitted either through WiFi or a serial port to the local ESP32 microcontroller. A programming cycle ends with the local microcontroller sending the bitstream to the FPRFS(s). The numbers shown on the corresponding segments in the software interface denote the order in the programming bitstream. The time for locally programming a 60-segment FPRFS is 25 us as shown in Supplementary Fig. 2, which corresponds to a maximum surface pattern refresh rate of 40,000 Hz. This time is proportional to the number of boards under configuration. The programming bitstream is measured by a RIGOL DS1104Z PLUS oscilloscope. During the automatic pattern-sweeping process, the software interface displays the pattern programmed in real time. The FPRFS application type, multi-device coordination, self-optimization, and self-adapting settings can be adjusted in detail as shown in Supplementary Fig. 19.

The pattern-sweeping algorithm substantially constructs the bitstream automatically following a certain rule. The 60-segment numbers are mapped to a new number order as shown in the bracket in the GUI for the convenience of coding adjacent segments. After setting the FPRFS application type and IO port positions, a brute force sweeping algorithm generates the FPRFS pattern accordingly and programs them sequentially to the FPRFS. Except for brute force sweeping, the user can save a range of FPRFS patterns as the preset for sweeping. The implemented FPRFS hardware and its complementary software leave the interface open for the development of more efficient pattern-sweeping algorithms.

## Data availability

The calculation, simulation, and experiment data that support the findings of this study are available in figshare with the identifier(s) https://doi.org/10.6084/m9.figshare.24973422.

## Code availability

The mathematical calculations and software interface algorithms are described in Methods and Supplementary Information. All relevant code is available at figshare with the identifier(s) https://doi.org/10.6084/m9.figshare.25478233.

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

## Acknowledgements

Grant FWO-Flanders project# G0B9821N: D.S.

## Author contributions

T.L. and E.S. conceived the FPRFS concept and wrote the manuscript. T.L. designed and fabricated the device. T.L. performed the theoretical analysis. T.L., Y.Y., and Y.L. participated in the HFSS simulation. T.L. performed the experiments. R.J.M. performed the antenna efficiency tests. R.R.U., D.S., and R.E. provided advice and discussed the results. All the authors reviewed and commented on the manuscript. E.S. supervised the research.

## Competing interests

The authors declare no competing interests.
