## [Peer Review File · Nature Communications]

REVIEWER COMMENTS

Reviewer #1 (Remarks to the Author):

1. An adaptive antenna consisting of a metallic grid connected by active devices is studied, fabricated, and experimentally tested in this paper. The shape and operating frequency of the antenna are reconfigurable via the use of a Field-Programmable Gate Array (FPGA). The reconfigurable transmission line segments can also help in antenna input impedance matching. The most significant and innovative finding is the use of an inverted polarity (IP) excitation scheme. It is demonstrated that this unbalanced antenna feeding scheme significantly improves the antenna's performance. The paper is of publishable quality in terms of innovation; however, its underlying operating mechanism still needs more clarifications for readers' better understanding.
2. Some of the equations from (1) to (14) can be placed in the Supplementary Materials for readers to focus on the key concepts in the paper.
3. The key contribution of the paper is explained between lines 351 to 369. However, the way it is presented is a bit hand waving. For example, the statements "The greatly increased return current on the feeder also explains the more omnidirectional H-plane radiation patterns and higher immunity to obstacles near the FPRFS", "we conclude that increased feeder return current leads to improved FPRFS antenna radiation performances", "However, to maintain the manipulability of FPRES antenna radiation characteristics, the return current radiation should not dominate." More explanation of the radiation mechanism should be given to demonstrate why the IP approach can enhance the radiation performance.
4. Antenna gain is typically compared with an isotropic radiator in dBi, not dB.
5. With the active devices' switching, the antenna's operating frequency is reconfigurable. Please also address the differences and advantages of the proposed approach with frequency reconfigurable antennas.
6. The radiation patterns shown in Fig. 4 show significant discrepancies between the simulated and measured results of the IP case, especially at the back radiation in Fig. 4(i), where it is more than 20 dB. How about the radiation efficiency of the measured and simulated cases under IP excitation?
7. In Fig. 4, the REF patch and the FPRFS patch have similar reflection coefficients. However, for their electric field cavity mode comparisons, the FPRFS patch has a smaller size compared with the REF one, but its TM₁₀ resonance point is lower. Please explain why.
8. In Fig. 5 m and n, we can see from both NIP cases the top side middle vertical obstacle has much more influence (pattern deterioration) on the antenna compared with the case when the bottom side middle vertical obstacle. Please explain why.
9. Only radiation efficiency is mentioned in Fig. 5. How about the total efficiency? Total efficiency is also

important in antenna design.

10. The scale on the vertical axis of Fig. 5 b-d, f-h, etc., is not correct.

11. As mentioned in the paper, the proposed design allows for optimal frequency of operation, polarization, etc. Does that mean the polarization can also be switched between linear and circular polarization within the variable frequency range? Similarly, Fig. 1 b illustrates that the smart FPRFS can provide beams pointing away from the broadside direction. The usefulness of the paper can be much enhanced if the authors can demonstrate that the proposed antenna can achieve the function illustrated in Fig. 1 b, even if just by simulation of switching on a set of particular active devices.

Reviewer #2 (Remarks to the Author):

The paper presents a concept called Field-Programmable Radio Frequency Surface (FPRFS) that enables the control of current flow patterns on the surface. This technology holds potential for achieving adaptable antennas and impedance matching networks. The study incorporates a combination of analytical calculations, simulations, and experimental demonstrations. While I found the paper interesting to read, it may not meet the publishing standards of Nature Communications. However, I recommend it as a suitable article for IEEE transactions.

As previously mentioned, the paper is well-organized and presents detailed results. However, one key aspect that lacks novelty is the limited exploration of new principles and mechanisms. The research content of this paper is more focused on technical issues, resulting in negligible innovation. Below are some suggestions for the authors to consider.

1) Firstly, the manuscript utilizes handheld device antennas as a conceptual illustration for the smart FPRFS, which I found intriguing due to its potential immediate applications. However, the size of the demonstrated sample, which has a dipole length (l_0) of 10 mm and an overall dimension of 85.5 mm x 85.5 mm, appears to be excessively large. This size is impractical for any handheld device, and the weight is not addressed in the manuscript. It would be beneficial to explore whether the design can be reduced in size while still maintaining a precise approximation of a small current dipole with a uniform constant current distribution.

2) Although the paper claims that the self-adaptive capability is achieved through a feedback control loop, the method for achieving adaptive control over the current flow pattern on the surface under varying environmental conditions remains unclear. For instance, it is necessary to clarify how feedback control technology is employed to configure the pattern along with the angle, as illustrated in Figure 6.

3) Building upon Comment 2, an important aspect that lacks sufficient information is the reconfiguration speed of the sample. It would be beneficial to address this concern by specifying the algorithms used for surface pattern sweeping to identify the optimal FPRFS pattern. Additionally, demonstrating the real-

time adjustment capability would enhance the overall evaluation.

4) Furthermore, considering the implementation of complex functions, the paper discusses the cascading of two FPRFSs. It raises an intuitive question: Is it possible to combine the two current patterns into a single FPRFS?

5) It would be beneficial to receive comments from the authors regarding the advantages and disadvantages of the current design compared to previous programmable metasurfaces.

6) Figure 4i-4j presents significant differences in the measured and simulated results for both NIP and IP, with deviations exceeding 25 dB at certain angles. It would be valuable to provide comments or explanations regarding these discrepancies.

7) The manuscript needs to be polished substantially. Some typos for example, "instead of", "single points", "proccs", "mathcing".

Reviewer #3 (Remarks to the Author):

Reviewer's Evaluation

In this paper, the author has presented an electronically reconfigurable antenna architecture referred to as Field-Programmable Radio Frequency Surface to implement different communication standards. This is achieved by manipulating the current flow on the surface, allowing for adaptable antennas and impedance matching networks. Additionally, by employing an asymmetric excitation of antennas on adjacent surfaces and dynamically controlling the return current on a segmented ground plane using PIN diode switches, a significant improvement in radiation efficiency is observed. The antenna's operational frequency range can be adjusted from 850 MHz to 3850 MHz, with a dynamic bandwidth of 3 GHz. To enhance adaptability, self-optimization algorithms are developed and integrated with the feedback hardware for both the FPRFS antenna (based on signal power feedback) and the FPRFS impedance matching network (IMN) (based on VSWR feedback). Changes in different RF environments trigger the algorithm to explore various FPRFS patterns (by modifying antenna geometry) and select the pattern with the highest gain in the desired direction. Altering antenna patterns also leads to corresponding adjustments in FPRFS IMN patterns, selecting the pattern with the lowest VSWR.

Major Comments

1. The reconfiguration of antenna structures by dynamically manipulating surface currents is a well-documented concept, and also the application of variously shaped metallic pixels, grid-lines and infinitesimal dipoles have been presented in the previous literature [manuscript references 23, 24]. Additionally, RF switches have been extensively utilized to alter the configuration of the radiating surfaces, leading to changes in surface currents distribution and consequently, radiation characteristics of the antennas. Also, reconfigurable multiple stub impedance matching networks using RF switches have been well-presented [M. Unlu et al., "A Reconfigurable RF MEMS Triple Stub Impedance Matching Network," in 2006 European Microwave Conference, 10-15 Sept. 2006, pp. 1370-1373]. Could the authors elucidate the primary conceptual innovation in their current work, apart from the automation of

the reconfiguration process?

2. In line 38, it is claimed that field programmability can improve polarization, whereas there is no mention of this in the entire paper. Please comment.
3. The conceptual representation in Figure 1b lacks sufficient explanation. A more detailed elaboration is required for the depiction of the operation of digital smart FPRFS in countering interferences and the incorporation of feedback from base stations.
4. There are two different setups for VSWR, and radiation pattern based self-optimisation respectively. Please elaborate and provide pictures for the complete operational setup for adaptive self-optimisation. Give comment on the practicality of having two different setups for providing self-optimisation function for a single device application.
5. In Figure 6h-g, self-adapting test for antenna and IMN are illustrated. Please provide pictures or video of the test for verification. Is there also a test involving wetting of IMN board instead of antenna?
6. The requirement of heavy and expensive equipment like RF signal generator and VNA in setups for adaptive self-optimisation makes the FPRFS an unsuitable candidate for practical user applications. Please comment on the practical utility of this design.

The reviewer has the following minor comments.

1. The mathematical derivation of radiation intensity based on infinitesimal dipoles is readily available in existing literature and appears unnecessary for inclusion such a detailed derivation in the manuscript.
2. In line 33, 34, please provide references for radiation efficiency comparison with literature.
3. Please explain why gain is not described in dBi throughout the paper?
4. Loss tangent for FR-4 substrate is mentioned as 0.0009, which is incorrect. [Line 518]

Response to Reviewers

We thank the anonymous reviewers for their insightful questions, comments, and suggestions. We believe that they consequently have significantly improved the quality of our manuscript.

Please find our response to each of their comments in the text below. We have also added line references to the manuscript and supplementary materials for their benefit

Reviewer 1

1. An adaptive antenna consisting of a metallic grid connected by active devices is studied, fabricated, and experimentally tested in this paper. The shape and operating frequency of the antenna are reconfigurable via the use of a Field-Programmable Gate Array (FPGA). The reconfigurable transmission line segments can also help in antenna input impedance matching. The most significant and innovative finding is the use of an inverted polarity (IP) excitation scheme. It is demonstrated that this unbalanced antenna feeding scheme significantly improves the antenna's performance. The paper is of publishable quality in terms of innovation; however, its underlying operating mechanism still needs more clarifications for readers' better understanding.

2. Some of the equations from (1) to (14) can be placed in the Supplementary Materials for readers to focus on the key concepts in the paper.

These equations have been moved as per the reviewers' suggestion.

(updated text appears between Line 78 and Line 122 in the Supplementary Information)

3. The key contribution of the paper is explained between lines 351 to 369. However, the way it is presented is a bit hand waving. For example, the statements "The greatly increased return current on the feeder also explains the more omnidirectional H-plane radiation patterns and higher immunity to obstacles near the FPRFS", "we conclude that increased feeder return current leads to improved FPRFS antenna radiation performances", "However, to maintain the manipulability of FPRES antenna radiation characteristics, the return current radiation should not dominate." More explanation of the radiation mechanism should be given to demonstrate why the IP approach can enhance the radiation performance.

Firstly, we demonstrated the enhanced return current on the feeder by inverting the feeder polarity. Secondly, unlike traditional coaxial feeders that utilize the outer metallic shield for grounding and shielding against electromagnetic radiation, the inverted polarity technique makes the feeder a part of the antenna radiator. Thirdly, we demonstrated this combined radiation is manipulable by programming our FPRFS as the main antenna structure. An important pre-requisite for this is that

neither the FPRFS antenna nor the feeder radiation dominates the total radiation. Finally, the automatic fast self-adjusting capability of our FPRFS is demonstrated to mitigate the side effects inherent in the unbalanced structure, including VSWR. A more detailed technical explanation with supporting simulation and experimental results is provided in the revised main text.

(updated text appears between Line 310 and Line 341 in the manuscript)

4. Antenna gain is typically compared with an isotropic radiator in dBi, not dB.

Thank you. The unit for antenna gain has been changed to dBi.

5. With the active devices' switching, the antenna's operating frequency is reconfigurable. Please also address the differences and advantages of the proposed approach with frequency reconfigurable antennas.

The FPRFS approach proposed in this paper fundamentally differs from frequency reconfigurable antennas in some very important ways. Notwithstanding the many benefits of frequency configurable antennas, the approach proposed here allows for additional degrees of freedom including flexibility in choosing antenna type, orientation, polarization, and impedance matching

Figure 1. (a) Geometry of the frequency-reconfigurable antenna¹. (b) Fabricated frequency-reconfigurable antenna. (c) Details of switch configuration.

networks. These can be configured to achieve overall desirable system attributes beyond the frequency of operation.

Other frequency reconfigurable antennas exhibit limited structural flexibility². An example is shown in **Figure 1**¹. Generally, the operating frequency of an antenna is closely tied to the RF current distribution. The number of current patterns of a reconfigurable antenna determines its frequency or operation, radiation pattern, and input impedance. Our FPRFS incorporates 60 elements, collectively creating an extensive array of distinct current paths, where each is added to the structure and contributes to radiation pattern, frequency of operation, and input impedance. The FPRFS concept is proposed to emulate arbitrary conductor shapes, allowing for increased flexibility. This approach results in higher frequency agility and a broader overall dynamic bandwidth.

6. The radiation patterns shown in Fig. 4 show significant discrepancies between the simulated and measured results of the IP case, especially at the back radiation in Fig. 4(i), where it is more than 20 dB. How about the radiation efficiency of the measured and simulated cases under IP excitation?

The discrepancies are mainly caused by the simulation model previously used, which consists of the FPRFS and the SMA feeder, without the extended feeding coaxial cable. The simulated and measured results for the NIP case show greater consistency because the return current on the NIP is small and its influence on the total radiation is negligible. For the IP case the increased return current due to the intentionally unbalanced feeding scheme also contributes to the measured radiation pattern. This was not captured in the simulation model but is observed in actual measurements. The old simulation model, new and updated simulation model and results are now provided in **Figure 2**.

The simulated and measured radiation efficiency results as a function of frequency are shown in **Figure 3**. The FPRFS patch with IP shows a significantly, better than ten-fold, greater efficiency than the NIP antenna in both simulated and measured results. The efficiency of the FPRFS antenna with IP is comparable to that of the conventional patch reference antenna in its principal operating frequency and surpasses that for the rest of the spectrum.

(updated figures for the radiation patterns appear in Fig. 5a-c in the manuscript; updated simulated and measured efficiency results appear in Fig. 5d and the text appears between Line 277 and Line 281 in the manuscript)

Figure 2. (a) Previous HFSS simulation model. (b) Modified HFSS simulation model. (c) Measured radiation patterns with previously simulated radiation pattern results. (d) Measured radiation patterns with updated simulated radiation pattern results.

7. In Fig. 4, the REF patch and the FPRFS patch have similar reflection coefficients. However, for their electric field cavity mode comparisons, the FPRFS patch has a smaller size compared with the REF one, but its TM₁₀ resonance point is lower. Please explain why.

The reason why the FPRFS emulated patch antenna has a smaller cavity size is because the PIN diode-based RF switches have an electrical length. The equivalent electrical length of the PIN diodes is a function of its operating frequency, its location in the active antenna pattern, the on/off

Figure 3. (a) Simulated radiation and total efficiency results of REF, NIP, IP 1×2-2-1 patch antennas as functions of frequency. (b) Measured radiation and total efficiency results of REF, NIP, IP 1×2-2-1 patch antennas as functions of frequency.

status of its adjacent segments, and the application (antenna or impedance matching networks). In this FPRFS emulated patch antenna scenario, the PIN diode’s equivalent electrical length is approximately 5.5 mm. Therefore, the equivalent patch antenna length is about 28 mm. The length of the conventional patch antenna shown in Fig. 4f,g,h is 26.28 mm. Therefore, although the FPRFS emulated patch antenna has a smaller physical size its electrical length is larger and comparable to the conventional patch antenna.

(updated text appears between Line 252 and Line 254 in the manuscript)

8. In Fig. 5 m and n, we can see from both NIP cases the top side middle vertical obstacle has much more influence (pattern deterioration) on the antenna compared with the case when the bottom side middle vertical obstacle. Please explain why.

In the revised manuscript, the previous Fig.5m,n are changed to Fig.6b,d. This phenomenon illustrates that for conventional NIP FPRFS antenna, energy is mostly radiated from the top side into the top hemisphere. When the obstacle is only attached to the top surface, significant pattern deterioration is observed. When the obstacle is only attached to the bottom surface, its influence is not as great. This illustrates that the radiation pattern of the NIP antenna is highly vulnerable to obstacles above it, whereas the IP antenna is less susceptible. The aim here is to illustrate that by using the IP feeding scheme, the radiation pattern deterioration is less pronounced.

(updated figures appear in Fig.6 and the updated text appears between Line 300 and Line 307 in the manuscript)

9. Only radiation efficiency is mentioned in Fig. 5. How about the total efficiency? Total efficiency is also important in antenna design.

The simulated and measured radiation and total efficiencies for FPRFS emulated $1 \times 2 - 2 - 1$ patch antenna with NIP, IP, and the conventional patch antenna reference are shown in **Figure 3**. The results are included in the main text. The radiation efficiency and total efficiency for the FPRFS IP antenna are significantly improved relative to those of the NIP antenna and are comparable to the conventional patch reference. Notably, the total efficiency remains consistently high across the FPRFS's dynamic bandwidth. These findings further underscore the efficacy of the IP scheme in producing productive reconfigurable antennas.

(updated figures appear in Fig.5d in the manuscript and in Supplementary Fig. 14 in the Supplementary Information; the updated text appears between 277 and 281 in the manuscript)

10. The scale on the vertical axis of Fig. 5 b-d, f-h, etc., is not correct.

Thank you. This is fixed accordingly.

(updated figures appear in Fig. 5f,h,j in the manuscript)

11. As mentioned in the paper, the proposed design allows for optimal frequency of operation, polarization, etc. Does that mean the polarization can also be switched between linear and circular polarization within the variable frequency range? Similarly, Fig. 1 b illustrates that the smart FPRFS can provide beams pointing away from the broadside direction. The usefulness of the paper can be much enhanced if the authors can demonstrate that the

proposed antenna can achieve the function illustrated in Fig. 1 b, even if just by simulation of switching on a set of particular active devices.

Polarization can also be reconfigured. Polarization modes including linear polarization, elliptical polarizations with different axial ratios and orientations, and circular polarization can be reconfigured on the FPRFS. There are a few different approaches to achieve reconfigurable polarization. These include:

a. Changing the FPRFS antenna feeding point: the positioning of the feeding point plays a crucial role in influencing the current distribution and electric field orientation across the antenna structure. This, in turn, dictates the characteristics of the electric field in the radiated electromagnetic wave,

Figure 4 (a) FPRFS patch antenna patterns. (b) The simulated and measured polarization results for the corresponding FPRFS antenna patterns.

ultimately determining the antenna's orientation. As the example shown in **Figure 4(a)**, FPRFS emulated 1×2 patch antennas with a length of 1 segment have different feeding positions. The simulation results in **Figure 4(b)** (column one) reveal far-field electric field vector trajectories above the FPRFS at the TM_{10} mode frequency, showcasing two symmetric elliptical trajectories and a linear trajectory. The gain results measured by a rotating linear polarized antenna are included in **Figure 4(b)** (column two). Likewise, for patch antenna patterns featuring multiple optional feeding positions, elliptical polarizations with varying axial ratios and orientations can be dynamically reconfigured. These reconfigurations are supported by both simulated and measured results.

b. Changing the FPRFS antenna structure: the FPRFS consists of many cruciform RF switching units, which can be used to switch the current path. We found that simply switching the current direction without modifying the current path length also achieves reconfigurable polarization. The results based on a group of fork antenna patterns are shown in **Figure 5**. The fork antennas under test consist of a 1-segment long branch and a 2-segment long branch but with different orientations from the feeding point. The current distribution on the fork antenna structure is configurable by adjusting these orientations. The FPRFS fork antenna patterns shown in **Figure 5(a)(b)** are simulated and measured at 2.45 GHz. Linear polarization and elliptical polarizations with different axial ratios and directions are configurable, as shown by the simulated and measured results in **Figure 5(c)(d)**.

Figure 5 (a),(b) FPRFS fork antenna patterns. (c),(d) The simulated and measured polarization results for the corresponding FPRFS antenna patterns.

c. Antenna array on the FPRFS: this approach on the FPRFS resembles a phased array antenna. Due to the flexibility of the proposed FPRFS, multiple identical antenna elements can be configured on it, each with a distinct excitation port and source. The ability to tune phase delays

Figure 6 (a) The FPRFS configuration of an antenna array with two identical 1×2 patch antenna elements. (b) The simulated polarization results of the FPRFS-based antenna array excited by two sources with identical amplitude but varying phase differences.

to each element allows for dynamic polarization reconfiguration. An example is shown in **Figure 6**. When two achiral 1×2 patch antenna elements are configured with separate feeding ports, the polarization is circular when the phase delay between these two excitation sources is orthogonal. When two chiral 1×2 patch antenna elements are configured and the delay between them in-phase or out-of-phase, the polarization is linear. The polarization sense (LHCP or RHCP) can be configured by selecting either a leading or lagging phase difference between two excitations. Furthermore, the axial ratio and polarization orientation of elliptical polarizations can be tuned by adjusting the source phase differences, splitting power strength to each excitation port, and configuring the shape of individual antenna elements on the FPRFS.

The proposed FPRFS can deliver frequency agility, radiation pattern reconfigurability, and polarization reconfigurability on a single device.

(updated text and figures appear between Line 231 and 266 in the Supplementary Information)

The FPRFS-based antenna demonstrates the capability to steer its beam in different directions through the reconfiguration of its active surface pattern. The corresponding results for IP cases are presented in Fig. 7c in the main text. For NIP cases, both the beam number and beam direction can

Figure 7 (a) FPRFS antenna patterns for different beam shapes. (b) S11 responses of these FPRFS antenna patterns. (c) The simulated 3D radiation patterns. (d) The simulated radiation patterns in H-plane.

be configured. Taking the example of the FPRFS-emulated 1×2 -2-1 patch antenna introduced in the main text, adding side branches, as depicted in **Figure 7a**, can modify the antenna radiation pattern beam properties. Simulated 3D and 2D H-plane radiation patterns show clear discrepancies in the antenna lobes as depicted in **Figure 7(c)(d)**.

Given that the current distribution on an antenna structure significantly influences its RF characteristics, encompassing frequency, radiation pattern, and polarization, the FPRFS, designed with arbitrary current control, enables reconfigurable antenna RF characteristics.

Reviewer 2

The paper presents a concept called Field-Programmable Radio Frequency Surface (FPRFS) that enables the control of current flow patterns on the surface. This technology holds potential for achieving adaptable antennas and impedance matching networks. The study incorporates a combination of analytical calculations, simulations, and experimental demonstrations. While I found the paper interesting to read, it may not meet the publishing standards of Nature Communications. However, I recommend it as a suitable article for IEEE transactions.

As previously mentioned, the paper is well-organized and presents detailed results. However, one key aspect that lacks novelty is the limited exploration of new principles and mechanisms. The research content of this paper is more focused on technical issues, resulting in negligible innovation. Below are some suggestions for the authors to consider.

1) Firstly, the manuscript utilizes handheld device antennas as a conceptual illustration for the smart FPRFS, which I found intriguing due to its potential immediate applications. However, the size of the demonstrated sample, which has a dipole length (l_0) of 10 mm and an overall dimension of 85.5 mm x 85.5 mm, appears to be excessively large. This size is impractical for any handheld device, and the weight is not addressed in the manuscript. It would be beneficial to explore whether the design can be reduced in size while still maintaining a precise approximation of a small current dipole with a uniform constant current distribution.

The FPRFS proposed in this work represents a concept that can be scaled to apply to various RF devices with different dimensions operating across different frequency ranges. Our FPRFS is designed and fabricated for operation between 850 MHz and 3850 MHz.

The dipole length (l_0) of 10 mm was chosen as it is greater than the equivalent electrical length of the PIN diode-based RF switches. The PIN diode has a physical length of less than 1 mm and an equivalent electrical length of 5-6 mm in the 5GHz range. The design of smaller and more compact appropriate switches will be required for smaller more compact FPRFS designs. Another issue limiting miniaturization will be the coupling between the adjacent dipole elements.

The weight of a single FPRFS is dominated by the weight of the PCB substrate. In the implementation of this paper, it is approximately 30 grams.

(updated text appears in Line 36 and between Line 41 and 46 in the Supplementary Information)

2) Although the paper claims that the self-adaptive capability is achieved through a feedback control loop, the method for achieving adaptive control over the current flow pattern on the

surface under varying environmental conditions remains unclear. For instance, it is necessary to clarify how feedback control technology is employed to configure the pattern along with the angle, as illustrated in Fig 6.

Experimentally, the setup is illustrated in **Figure 8**. The RF signal generator provides the excitation to the TX antenna. The energy received by the RX antenna is converted into a DC voltage by the RF power meter. This measured DC voltage serves as the signal for optimizing the TX antenna radiation pattern characteristics.

In the experimental setup, the FPRFS is rotated to emulate differences in the orientation between the FPRFS TX antenna and the target receiver RX. For any relative position, the system sweeps continuously to configure all possible surface patterns on the FPRFS (e.g., from pattern No. 1 to pattern No. 80). This can be achieved very quickly. After a full sweep cycle, the algorithm sorts the feedback DC voltages, corresponding to received power, and identifies the FPRFS antenna pattern that achieves the greatest received signal strength at the RX. As the relative position changes through rotation, this sweeping and conditioning process repeats, and all FPRFS antenna patterns, along with their corresponding measured radiation patterns, are depicted in the main text Fig. 7b,c. Similarly, for VSWR optimization, the feedback signal measured the locally reflected power at the FPRFS TX side, measured at the directional couplers reflected port.

Figure 8 (a)(b) Experimental setup for self-optimizing FPRFS antenna (TX) gain and radiation pattern at the target location (RX).

In practical scenarios, the feedback control process described is used in adaptive wireless communications, including beamforming antennas³ and cognitive radios⁴, and also implemented in IEEE standards⁵. As Fig. 1b illustrates, our FPRFS is designed for the adaptability of far-field radiation characteristics and local impedance matching. Optimizing far-field radiation characteristics aligns with beamforming antenna systems. The overall process is as follows:

1. **Channel estimation:** Initiated by transmitting known pilot signals or training sequences, the receiver analyzes and determines key factors such as channel gain, phase, and interference. The outcomes of channel estimation serve as feedback information transmitted back to the FPRFS TX antenna.

2. **Feedback transmission:** The quantized and encoded feedback data are packaged into frames and transmitted from the receiver back to the transmitter.

5. **Processing at the transmitter:** The FPRFS-based software at the transmitter side recovers the feedback information and stores it for further analysis.

6. **Decision-making:** after a complete sweeping cycle, all feedback information is processed, and the software identifies the FPRFS pattern that achieves the transmission channel with the highest signal-to-noise ratio (SNR).

7. **Iterative process:** Multiple sweeping cycles can be conducted to enhance result reliability. In a self-adapting system, continuous monitoring of CSI information occurs at specified intervals. If the transmission channel degrades, the entire optimization process is initiated and iterated. This iterative feedback loop ensures ongoing adaptation, allowing the system to respond dynamically to changes in the transmission environment.

The optimization feature based on VSWR and matching characteristics, presented as a significant novelty in this work, is seldom realized in existing adaptive wireless communication systems. Its overall process unfolds as follows:

1. **VSWR estimation:** Initiated by transmitting known pilot signals or training sequences, the directional coupler measures the reflected power to estimate the VSWR.

2. **Feedback generation and conversion:** The reflected RF power is converted into a signal that a microcontroller can read, serving as feedback information for VSWR and input impedance. This can be implemented, like in this work, with a directional coupler and RF diode circuit.

3. **Processing at the transmitter:** The FPRFS-based software stores the data for further analysis.

4. **Decision-making:** After a complete sweep, the FPRFS impedance matching pattern that achieves the lowest VSWR is chosen.

5. **Iterative process:** Multiple sweeping cycles can be conducted to enhance result reliability. In a self-adapting system, continuous monitoring of VSWR occurs at specified intervals. If the VSWR increases, the entire optimization process is initiated and iterated.

A schematic diagram and a flow chart illustrating self-optimization and self-adaptiveness are included in the main text Fig. 8a,b. In practical applications, base stations often act as the RX, offering a larger network range and robust processing power. Consequently, the base station takes on the task of processing feedback and making decisions. It efficiently sends back only the optimal FPRFS pattern index after a sweeping cycle, ensuring a quicker overall response and reducing the local processing workload. When dealing with multiple feedback sources, as illustrated in the main text Fig. 1b (showcasing feedback for VSWR and far-field gains at three base stations), intelligent

algorithms will need to be developed to determine the weight allocated to each feedback source and make a comprehensive optimal decision.

(text and figures updated between Line 393 and Line 399 in the manuscript)

3) Building upon Comment 2, an important aspect that lacks sufficient information is the reconfiguration speed of the sample. It would be beneficial to address this concern by specifying the algorithms used for surface pattern sweeping to identify the optimal FPRFS pattern. Additionally, demonstrating the real-time adjustment capability would enhance the overall evaluation.

The details of the FPRFS reconfiguration speed are illustrated in the original version of Supplementary Information between Line 41 and 49 (*in the updated version of Supplementary Information between Line 47 and 53*). The reconfiguration speed is controlled by three system components.

The first part is the FPRFS hardware reconfiguration speed. This speed is calculated by the time it takes for the FPRFS biasing circuits to fully configure all the RF switches from a programming bitstream. As shown in the Supplementary Information, it takes about 25 us to configure the pattern on the prototype FPRFS denoting that the FPRFS has a pattern refresh rate of about 40,000 Hz. In the current implementation, the reconfiguration speed is limited by the communication speed with the configuration systems (computer speed). The lower bound fundamental limit is dictated by the number of switches that need to be configured and the time to bias the switches.

The second part is the system coordination speed when the feedback control loop is applied. For self-optimization and adaptation, the FPRFS itself can be reconfigured very quickly. The controlling software must receive feedback from the remote sensing unit before configuring the next pattern to the FPRFS. This is controlled by the communications protocol and the probability that packets containing feedback measurements are correctly received. Timeouts must also need to be considered (lost packet or if the configuration leads to an antenna configuration where coding gain is insufficient to decode the packet).

At this moment in time, the search process to obtain the optimal pattern is brute force. In future work, algorithms must be developed to optimize the radiation pattern search. This will be especially important as the FPRS becomes larger with more configuration capability.

The real-time capability is demonstrated from four aspects including far-field radiation pattern, near-field loading effects, physical structure damage, and environmental factors. The corresponding experimental schematics are updated in the main text and the demonstration videos are updated as Supplementary Movies 5-10.

(demonstration videos are updated as Supplementary Movie 5-10)

4) Furthermore, considering the implementation of complex functions, the paper discusses the cascading of two FPRFSs. It raises an intuitive question: Is it possible to combine the two current patterns into a single FPRFS?

Figure 9 (a) A 5th order band stop filter pattern on a single FPRFS and a 10th order band stop filter pattern on a combined FPRFS. (b) Theoretical S21 response results of the 5th order and 10th-order filters. (c) The 5x2 patch antenna pattern on a combined FPRFS extended with SMA connectors. (d) The 5x2 patch antenna pattern on a combined FPRFS extended with direct mounting. (e) The simulated S11 responses for the 5x2 patch antenna pattern extended with SMA connectors. (f) The simulated S11 responses for the 5x2 patch antenna pattern extended with direct mount.

We thank the reviewer for the excellent suggestion and insight the question has provided.

Yes. Presently, the FPRFS is designed with SMA connectors surrounding the device, allowing for cascading of multiple FPRFS units.

In applications involving transmission lines, such as impedance matching networks, RF filters, power dividers, and combiners, integrating two current patterns into a single FPRFS is advantageous as it provides greater flexibility. One example of using this FPRFS as a reconfigurable RF filter is shown in **Figure 9(a)(b)**. For microstrip line RF filters, increasing the number of side stubs enhances the filter order, enabling steeper passband-stopband transition. While a single FPRFS (as currently designed and implemented) achieves a maximum filter order of 5th, combining two FPRFS units can achieve a 10th filter. This combination approach also opens up possibilities for more intricate impedance matching network (IMN) patterns.

A more complex antenna would be achievable by cascading FPRS. However, careful placement (symmetric) of elements (dipoles) and connectors would probably be required. Specifically, the distance between elements and the electrical length of the connectors and cables must be taken into account otherwise a discontinuous antenna structure may result, as illustrated in **Figure 9(c)**, where a single antenna is implemented across multiple FPRFS. If an antenna is contained within a single FPRFS, multiple FPRFS units can be connected directly, as depicted in **Figure 9(d)**, and this allows the realization of reconfigurable antennas on a larger single FPRFS. This configuration offers greater flexibility with more configurable current dipole pixels and higher pattern resolution. The multi-FPRFS-emulated 5×2 antenna, as shown, exhibits a lower principal operating frequency as the patch antenna length increases. This is caused by the RF chokes on the FPRFS, as illustrated in Fig. 4d and described in the main text.

5) It would be beneficial to receive comments from the authors regarding the advantages and disadvantages of the current design compared to previous programmable metasurfaces.

Metasurfaces are metamaterials, which are usually composed of ultrathin planar meta-atoms (discrete subwavelength structures) with predesigned electromagnetic responses⁶. These structures are engineered to manipulate the amplitude, phase, polarization, and other parameters of incident electromagnetic waves⁷. Metasurfaces receive externally incident EM waves and alter their propagation properties, including refraction, reflection, scattering, transmission, and absorption. *An important point is that the signal to be manipulated has been launched into space and is propagating in space or a waveguide (e.g. rectangular waveguide). Another antenna or transition (coaxial to waveguide) is required.* Here the FPRFS acts as the radiator, no other device is required, another critical distinction is that the FPRFS can also be used to both filter (remove unwanted spectral content) and impedance match.

Different metasurface types cater to various functions, such as frequency-selective, high-impedance, perfectly absorbing, and wave-front shaping metasurfaces. A metasurface modifies the

Figure 10 (a) A multi-layer polarization-modulated metasurface design¹². (b) An Optically Tunable Metasurfaces realized with optically sensitive polymers for dynamic polarization manipulation of visible light¹³. (c) Gate-controlled diffractions of light at a metal/insulator/metal metadvice with ITO as its spacer, under appropriately chosen electric gating¹⁴. (d) Schematic of a multifunctional polarization converter¹⁵.

output EM waves in response to its incident EM waves, as illustrated in **Figure 10**. A metasurface modifies the output EM waves in response to its incident EM waves, as illustrated in **Figure 10a-d**.

Antenna power transmitting efficiency and radiation pattern are not key design considerations for metasurfaces. This is because there is another component that has launched the signal that is now propagating in space. In contrast, these are critical design concerns for the FPRFS. Although the FPRFS can be used in metasurface applications, the FPRFS, as disclosed here importantly provides for the efficient control of the flow of current injected onto its surface to implement structures (impedance matching structures and filters) and antennas with high efficiency.

When comparing with previous reconfigurable antennas, the advantages of our work include our response to Comment 5 from Reviewer 1 plus:

Multifunctionality: the pixelated design enables the emulation of arbitrary current flow patterns. The abundance of IO ports further expands the scope for reconfigurable RF applications, encompassing impedance matching networks, RF filters, power dividers or combiners, and the combination of these functionalities—all achievable on a single device.

6) Figure 4i-4j presents significant differences in the measured and simulated results for both NIP and IP, with deviations exceeding 25 dB at certain angles. It would be valuable to provide comments or explanations regarding these discrepancies.

This has been addressed in response to Comment 6 from Reviewer 1.

(updated figures for the radiation patterns appear in Fig. 5a-c in the manuscript)

7) The manuscript needs be polished substantially. Some typo for example, “isntead of”, “single points”, “procss”, “mathcing”.

These have been fixed accordingly.

Reviewer 3

In this paper, the author has presented an electronically reconfigurable antenna architecture referred to as Field-Programmable Radio Frequency Surface to implement different communication standards. This is achieved by manipulating the current flow on the surface, allowing for adaptable antennas and impedance matching networks. Additionally, by employing an asymmetric excitation of antennas on adjacent surfaces and dynamically controlling the return current on a segmented ground plane using PIN diode switches, a significant improvement in radiation efficiency is observed. The antenna's operational frequency range can be adjusted from 850 MHz to 3850 MHz, with a dynamic bandwidth of 3 GHz. To enhance adaptability, self-optimization algorithms are developed and integrated with the feedback hardware for both the FPRFS antenna (based on signal power feedback) and the FPRFS impedance matching network (IMN) (based on VSWR feedback). Changes in different RF environments trigger the algorithm to explore various FPRFS patterns (by modifying antenna geometry) and select the pattern with the highest gain in the desired direction. Altering antenna patterns also leads to corresponding adjustments in FPRFS IMN patterns, selecting the pattern with the lowest VSWR.

Major concerns

1. The reconfiguration of antenna structures by dynamically manipulating surface currents is a well-documented concept, and also the application of variously shaped metallic pixels, grid-lines and infinitesimal dipoles have been presented in the previous literature [manuscript references 23, 24]. Additionally, RF switches have been extensively utilized to alter the configuration of the radiating surfaces, leading to changes in surface currents distribution and consequently, radiation characteristics of the antennas. Also, reconfigurable multiple stub impedance matching networks using RF switches have been well-presented [M. Unlu et al., "A Reconfigurable RF MEMS Triple Stub Impedance Matching Network," in 2006 European Microwave Conference, 10-15 Sept. 2006, pp. 1370-1373]. Could the authors elucidate the primary conceptual innovation in their current work, apart from the automation of the reconfiguration process?

This work proposes numerous innovations that potentially make reconfigurable antennas productive.

- a. Our first conceptual innovation is to address the major challenge of reconfigurable antennas, namely the loss of radiation efficiency due to lossy RF switches⁸. We propose utilizing an unbalanced antenna feeding approach where we excite the continuous metal top layer and control the imaged current on the (defective) ground plane. Here the defective ground plane is the pixelated surface where the current is controlled by lossy RF switches. We show that by using this approach, the antenna performance is only weakly dependent on the RF switch loss. *This approach is a major advance to the field as it breaks the conventional relationship between increased power loss and the drastic reduction in efficiency as the number of RF switches, enabling the use of a larger number of RF*

switching elements for improved replication of the desired antenna pattern. Researchers have devoted significant effort to overcome this problem by focusing on building very high-performance (usually mems-based) switches

- b. Another key innovation is introducing a universal pixelated design, providing a higher degree of flexibility in reconfiguring structures. Unlike previous works that reported only one RF application with reconfigurability using uniquely designed structural elements, our universal pixelated design allows for the implementation of multiple different RF devices (antennas, impedance matching networks, and RF filters) with characteristics reconfigurability on a single FPRFS device. ***Our approach enhances flexibility and improves the ability of the system to adapt to environmental conditions. The implementer can simultaneously implement an antenna that maximizes the radiation characteristics, achieves acceptable impedance matching, and rejects unwanted receive signals or avoids out-of-band emissions. These are all critical attributes for true adaptive systems.*** Although [manuscript references 23, 24] have reported pixelated reconfigurable antennas, they lack a scalable analytical model for theoretical support. Neither addresses the input impedance mismatch as the antenna configuration changes, and neither discusses the possibility of utilizing their devices for applications beyond antennas or any adaptive control capability. Our innovations, addressing these research gaps, broaden the scope of applications and enhance the adaptability of FPRFS in diverse scenarios.
- c. As mentioned, our proposed approach also allows for implementing Antennas and RF filters, in addition to impedance-matching networks. Front-end filters are critical to remove the effects of undesirable interferers from saturating the Low Noise Amplifiers or out-of-band emission due to power amplifier nonlinearities. The work presented here is a major advance on the work of Unlu *et al.* For completeness, Unlu *et al.*'s work is now referenced.

(the Unlu et al.'s paper is referenced in Line 172 in the manuscript)

2. In line 38, it is claimed that field programmability can improve polarization, whereas there is no mention of this in the entire paper. Please comment.

The response is as provided in Comment 11 to Reviewer 1.

(updated text and figures appear between Line 232 and 266 in the Supplementary Information)

3. The conceptual representation in Figure 1b lacks sufficient explanation. A more detailed elaboration is required for the depiction of the operation of digital smart FPRFS in countering interferences and the incorporation of feedback from base stations.

Our response is provided in 2 to Reviewer 2.

(text and figures updated between Line 393 and Line 399 in the manuscript)

4. There are two different setups for VSWR, and radiation pattern based self-optimisation respectively. Please elaborate and provide pictures for the complete operational setup for adaptive self-optimisation. Give comment on the practicality of having two different setups for providing self-optimisation function for a single device application.

For far-field radiation characteristics self-optimization, the applied RF meter-based setup is symbolically shown in the main text Fig. 7a and the experimental setup is shown in **Figure 11(a)(b)**. The TX source and RF terminal outside the anechoic chamber are connected to the TX FPRFS

Figure 11 (a)(b) Experimental setup for self-optimizing FPRFS antenna (TX) gain and radiation pattern at the target location (RX). (c) Experimental setup 1 for self-optimizing VSWR using a VNA. (d) Experimental setup 2 for self-optimizing VSWR using a signal generator and an RF meter.

antenna with coaxial cables. The rotation of the FPRFS simulates the dynamic change in relative positions between the FPRFS antenna and the target receiver. For VSWR self-optimization, setup 1 based on a VNA is symbolically shown in the main text Fig. 7d, and the experimental setup is shown in **Figure 11(c)**. For VSWR self-adapting tests, setup 2 based on a signal generator and an RF meter is symbolically shown in the main text as Fig. 8c and the experimental setup is shown in **Figure 11(d)**.

The radiation pattern and impedance matching network optimization are interdependent. In this experiment, an RF signal generator is used to simulate the transmitter and provide a signal to the antenna. One RF meter is attached to the coupling port of a directional coupler and measures the locally reflected power from the transmit antenna port. The second power meter is attached to the receive antenna. The optimization is, and the impedance matching networks are determined (or it can be stored) that meets the desired performance goal. The algorithm used is as follows. A transmit antenna pattern is first chosen and the impedance matching networks are determined (or can be stored) that meets the desired performance goal (ie -10dB or is the best one). The transmitter then transmits on this configuration and the receiver records the received power. The transmitter needs to keep track of the impedance matching network for each antenna configuration. The FPRFSs system at a target frequency of 2.4 GHz for both setups are attached as Supplementary Video 3 and Supplementary Video 4 respectively.

As discussed in our response to question 3, imagine a part of the FPRFS is programmed to do impedance matching and a part to work as an antenna, as depicted in Fig. 1b. When the self-optimization process starts, on the VSWR optimization side, the aim is to minimize the locally coupled reflected power thus to maximize the incident power to the FPRFS antenna part. On the radiation gain optimization side, the aim is to maximize the SNR at the target base stations, which relies on the channel estimation process generating the CSI and sending it back to the transmitter.

(updated text and figures appear between Line 355 and 392 in the Supplementary Information)

5. In Figure 6h-g, self-adapting test for antenna and IMN are illustrated. Please provide pictures or video of the test for verification. Is there also a test involving wetting of IMN board instead of antenna?

The real-time adjustment capability is demonstrated from four aspects including far-field radiation pattern, near-field loading effects, physical structure damage, and environmental factors. The corresponding results and explanations are updated in the main text and the demonstration videos are updated as Supplementary Movie 5-10.

Yes, a test involving wetting of IMN board instead of antenna is carried out accordingly and the video is updated as Supplementary Movie 10.

(demonstration videos are updated as Supplementary Movie 5-10)

6. The requirement of heavy and expensive equipment like RF signal generator and VNA in setups for adaptive self-optimisation makes the FPRFS an unsuitable candidate for practical user applications. Please comment on the practical utility of this design.

The VNA and RF signal generators are used only for testing convenience and are not required for FPRFS implementation. Laboratory test equipment for concept validation and verification is used universally in concept validation.

A VNA is not required, but notwithstanding this fact, VNA implemented as a single integrated circuit is available including the ADL 5960 chipset that is capable of analyzing 10 MHz to 20 GHz. In practical scenarios, electronic transmit signals would come directly from the transmitter. The directional coupler for VSWR can be produced very compactly on the transceiver PCB. Integrated single-chip transceivers implement power-determination circuitry. This would be implemented on both the transmit and the receive.

Minor concerns

1. The mathematical derivation of radiation intensity based on infinitesimal dipoles is readily available in existing literature and appears unnecessary for inclusion such a detailed derivation in the manuscript.

This is fixed according to the suggestion.

(updated text appears between Line 78 and Line 122 in the Supplementary Information)

2. In line 33, 34, please provide references for radiation efficiency comparison with literature.

The radiation efficiency comparison is derived from the results presented in Fig 5a-1. We have also measured, and now report the measured efficiency experimental results for the FPRFS-implemented antenna. It's worth noting that the FPRFS implement IP antenna radiation efficiency is close to traditional patch antennas^{8,9,10,11}.

3. Please explain why gain is not described in dBi throughout the paper?

Thank you. The unit is fixed to be dBi.

4. Loss tangent for FR-4 substrate is mentioned as 0.0009, which is incorrect. [Line 518]

Thank you. The correct value of 0.02 is now stated.

(updated in Line 502 in the manuscript)

References

1. Majid, H. A. *et al.* Frequency-Reconfigurable Microstrip Patch-Slot Antenna. *12*, 218–220 (2013).
2. Abutarboush, H. F. & Shamim, A. A Reconfigurable Inkjet-Printed Antenna on Paper Substrate for Wireless Applications. *IEEE Antennas Wirel. Propag. Lett.* *17*, 1648–1651 (2018).
3. Kassir, H. Al *et al.* A Review of the State of the Art and Future Challenges of Deep Learning-Based Beamforming. *IEEE Access* *10*, 80869–80882 (2022).
4. Walko, J. Cognitive radio. *IEE Rev.* *51*, 34–37 (2005).
5. IEEE. Part 11: Wireless LAN Medium Access Control (MAC) and Physical Layer (PHY) Specifications. *IEEE Std 802.11-2012 (Revision IEEE Std 802.11-2007)* 2012, 1–2695 (2012).
6. He, Q. Tunable / Reconfigurable Metasurfaces : Physics and Applications. 2019, (2019).
7. Zahra, S. *et al.* Electromagnetic Metasurfaces and Reconfigurable Metasurfaces: A Review. *Front. Phys.* *8*, 1–16 (2021).
8. Sheta, A. F. & Mahmoud, S. F. A widely tunable compact patch antenna. *IEEE Antennas Wirel. Propag. Lett.* *7*, 40–42 (2008).
9. Kang, S. H. & Jung, C. W. Transparent Patch Antenna Using Metal Mesh. *IEEE Trans. Antennas Propag.* *66*, 2095–2100 (2018).
10. Khan, M. U., Sharawi, M. S. & Mittra, R. Microstrip patch antenna miniaturisation techniques: A review. *IET Microwaves, Antennas Propag.* *9*, 913–922 (2015).
11. Papapolymerou, L., Drayton, R. F. & Katehi, L. P. B. Micromachined patch antennas. *IEEE Trans. Antennas Propag.* *46*, 275–283 (1998).
12. Pfeiffer, C., Zhang, C., Ray, V., Guo, L. J. & Grbic, A. High performance bianisotropic metasurfaces: Asymmetric transmission of light. *Phys. Rev. Lett.* *113*, (2014).
13. Ren, M. X. *et al.* Reconfigurable metasurfaces that enable light polarization control by light. *Light Sci. Appl.* *6*, e16254-5 (2017).
14. Huang, Y.-W. W. *et al.* Gate-Tunable Conducting Oxide Metasurfaces - Supplemental Material. *Nano Lett.* *16*, 5319–5325 (2016).

15. Wu, P. C. *et al.* Broadband Wide-Angle Multifunctional Polarization Converter via Liquid-Metal-Based Metasurface. *Adv. Opt. Mater.* 5, 1–7 (2017).

REVIEWER COMMENTS

Reviewer #1 (Remarks to the Author):

As stated in the original review, the most significant and innovative finding is the use of an inverted polarity (IP) excitation scheme. It argues that through this IP excitation, the increase in the number of diodes would not substantially increase the insertion loss, and this unbalanced antenna feeding scheme significantly improves the antenna's performance. Several comments were raised in the original submission, and the authors are commended for providing in-depth responses in the revised submission. Most of the responses are satisfactory, but further clarification may be needed for a couple of the comments.

Non-essential equations have been moved to the Supplementary Information.

The benefits of using an IP excitation scheme can be further explained/clarified. In Supplementary Note 9, Fig. 11, for the PIN diode characterization, the center pin of the SMA connector is connected to the transmission line. In the FPRES, the center pin is connected to the ground plane, which is the NIP mode. Would there be any difference in the results and conclusion? Also stated there is the average power consumption of an ON-state PIN diode is about 3 mW, and based on the DC biasing IV curve, the minimum power consumption is 0.53 mW. Again, in Supplementary Note 10, the cruciform RF switching unit shown in Fig. 12, the center pins are connected to the transmission line, not the ground plane. Would there be any difference in the results? It would be good if the authors could have some estimation of the total power dissipation due to the diodes in the IP and NIP excitations to substantiate the innovation of the IP excitation. In Supplementary Note 11, the measured efficiency shows that the IP excitation is better than that of the NIP and reference design. However, the simulated and measured efficiencies are quite different. In particular, in Fig. 14b, the simulated efficiency for (2x2-2-1) at around 1.5GHz is an efficiency dip, but that of the measurement is a peak. It would be beneficial to discuss how the efficiency is measured briefly. Simply saying that the cable lengths are different in the simulation and measurement may not be sufficient to explain the difference.

The unit of the antenna gain is corrected.

The advantage of using the proposed approach to tune the frequency is fine, but it may also change the radiation characteristics, such as radiation pattern.

The simulated and measured radiation patterns agree better due to a more accurate simulation model.

The resonant frequency is lowered due to the parasitic effects of the diodes.

The radiation of the NIP case is higher in the upper hemisphere than in the lower hemisphere. The authors explained the difference in loading effects satisfactorily.

The authors now also provided the total efficiency of the FPRFS. It would be good if the authors could briefly explain how the antenna efficiency is measured.

The scale on the vertical axis of Fig. 5 b-d, f-h, etc., has been fixed.

The authors have successfully illustrated various FRPFS performances under different configurations.

Reviewer #2 (Remarks to the Author):

Upon thorough review of the authors' response from the previous round of review and the revised manuscript, I am pleased to acknowledge the considerable efforts made by the authors in addressing the raised comments. The changes implemented have significantly enhanced and clarified the paper. However, before any potential recommendation can be made, there are still critical issues that necessitate consideration.

1. The authors have discussed the primary innovation achieved through the proposed FRPFS device in their current work. Nonetheless, in order to emphasize the enhancement and merit of the concept put forth in this study, it is imperative to undertake quantitative comparisons with prior research or state-of-the-art devices. This approach will enable a quantitative assessment of the advantages, such as the improved radiation efficiency/gain achieved by the proposed method. Such comparisons would be conducive to elucidating the authors' proposition.

2. The authors have clarified the optimization process in their response letter, highlighting it as a procedure of searching for optimal patterns and parameters within the parameter space. This method determines the optimal stimulation pattern by iteratively exploring all predetermined possible situations and selecting the one that enhances performance the most based on the signal received by the receiver, the base station in a conception. While this is a simple and direct method, it would be interesting to explore whether specific optimization algorithms, such as Artificial Neural Networks (ANN), can be implemented in this process.

3. In the response regarding the self-adaptive capability, the authors seem to determine the optimization features, in turn, based on the far-field radiation characteristics with the highest SNR, as well as the impedance matching pattern that achieves the lowest VSWR. What they do is iteratively check each pattern in a loop to determine if it is an option with the highest SNR, and repeat the process to search for the pattern with the lowest VSWR. This raises the question of how the optimal combination of parameters can be obtained through sequential searching steps. In reality, this constitutes a multi-objective optimization design, therefore, it would be beneficial to explain how the device is optimized by employing different objective functions in the proposed processes.

4. In response to comment 6 from reviewer #3, the authors claim that the VNA in the current experimental setup can be replaced by the ADL 5960 in practical scenarios. While this may be true from a functionality viewpoint, the use of integrated vector network analyzers such as ADL 5960 in mobile devices may not be economically practical for consumer electronics, including smartphones, due to their high cost. It would be helpful if the authors could propose more practical solutions for such scenarios.

Reviewer #3 (Remarks to the Author):

Authors replied to all the raised comments in the satisfactory manner.

Response to Reviewers

We thank the anonymous reviewers for their second review, their further insightful questions, comments, and suggestions. We believe that they consequently have significantly improved the quality of our manuscript.

Please find our response to each of their comments in the text below. We have also added line references to the manuscript and supplementary materials for their benefit

Reviewer 1

Non-essential equations have been moved to the Supplementary Information.

The unit of the antenna gain is corrected.

The simulated and measured radiation patterns agree better due to a more accurate simulation model.

The resonant frequency is lowered due to the parasitic effects of the diodes.

The radiation of the NIP case is higher in the upper hemisphere than in the lower hemisphere. The authors explained the difference in loading effects satisfactorily.

The scale on the vertical axis of Fig. 5 b-d, f-h, etc., has been fixed.

The authors have successfully illustrated various FRPFS performances under different configurations.

1. The benefits of using an IP excitation scheme can be further explained/clarified. In Supplementary Note 9, Fig. 11, for the PIN diode characterization, the center pin of the SMA connector is connected to the transmission line. In the FPRES, the center pin is connected to the ground plane, which is the NIP mode. Would there be any difference in the results and conclusion?

We have undertaken further studies with simulated and measured results showing the on/off behavior of the PIN diode. The results show an increase of about 2 dB in the insertion loss, and this is attributed to the presence of a segmented ground plane over which the transmission line travels. The simulated and measured PIN diode characterization results comparing with IP feeders and NIP feeders are shown in **Figure 1** and the corresponding results for RF switching unit are shown in **Figure 2**.

PIN diode characterization in NIP configuration is commonly used for testing. There the ON-state insertion loss and OFF-state isolation are determined. The fundamental goal of the NIP test is to assess the power transmitted through the waveguide, aiming to maximise the power transmitted in the ON-state whilst minimizing the power transmitted in the OFF-state. A solid ground helps minimize power loss in the ON-state. It then follows that this application finds IP feeders with the segmented lossy ground undesirable.

Figure 1 | PIN diode characterization results comparison. **a** The simulated and measured ON/OFF behaviors of the single-PIN diode RF switch with NIP feeders. **b** The simulated and measured ON/OFF behaviors of the double-PIN diode RF switch with NIP feeders. **c** The simulated and measured ON/OFF behaviors of the single-PIN diode RF switch with IP feeders. **d** The simulated and measured ON/OFF behaviors of the double-PIN diode RF switch with NIP feeders.

However, our design goal is to increase the radiation efficiency. The improved antenna radiation performance is attributed to increased return current for the antenna application. Although the IP feeder configuration makes the ground plane more lossy, it also increases return current and improves radiation performance. The relatively high OFF-state isolation is maintained in IP cases, which is critical for confining the energy within the programmed conductive pattern cavity, thus maintaining the IP FPRFS antenna's RF characteristic tunability.

(updated text appears between Line 308 and Line 325 and the added figures appear in Supplementary Fig. 12 in the Supplementary Information)

Also stated there is the average power consumption of an ON-state PIN diode is about 3 mW, and based on the DC biasing IV curve, the minimum power consumption is 0.53 mW.

Figure 2 | RF switching unit characterization results with IP feeders. **a** Cruciform RF switching unit configured to be all OFF. **b** Cruciform RF switching unit configured as an open end with one segment being biased to VCC (red part denoting the active RF current path). **c** Cruciform RF switching unit configured as an RF current router with two segments being biased to VCC. **d** Cruciform RF switching unit configured as an RF current divider or combiner with three segments being biased to VCC. **e** Cruciform RF switching unit configured as an RF current divider or combiner with four segments being biased to VCC.

The minimum power consumption of 0.53 mW is achieved at a PIN diode bias of 0.8 V. The power consumption increases to 3 mW when the bias is increased to 0.88 V. This 0.88 V is the 0.8 V minimum turn-on voltage plus a small 0.08 V margin to contend with any unaccounted voltage drops that may occur at any PIN diode and is applied for all PIN diodes on the FPRFS in all experiments. This is to ensure consistent and reproducible results. Supplementary Fig. 11d,f, show the results as a function of the bias voltage.

(updated text appears between Line 290 and Line 293 in the Supplementary Information)

Again, in Supplementary Note 10, the cruciform RF switching unit shown in Fig. 12, the center pins are connected to the transmission line, not the ground plane. Would there be any difference in the results?

Similarly, this is discussed in the response to Comment 1 and the simulated and measured PIN RF switching unit characterization results with IP feeders are shown in **Figure 2**.

(updated text appears between Line 351 and Line 353 and the added figures appear in Supplementary Fig. 15 in the Supplementary Information)

It would be good if the authors could have some estimation of the total power dissipation due to the diodes in the IP and NIP excitations to substantiate the innovation of the IP excitation.

As discussed previously, the feeder configuration (NIP or IP) would not affect the biasing status of the PIN diodes. The total power dissipation of the FPRFS only depends on DC biasing voltage and the number of activated segmented pixels. Discussion is provided in our response to the reviewer on the average power.

(updated text appears between Line 309 and Line 311 in the Supplementary Information)

In Supplementary Note 11, the measured efficiency shows that the IP excitation is better than that of the NIP and reference design. However, the simulated and measured efficiencies are quite different. In particular, in Fig. 14b, the simulated efficiency for (2x2-2-1) at around 1.5GHz is an efficiency dip, but that of the measurement is a peak.

Thank you. The two figures on the third and fourth rows should be exchanged. The corrected figure is now shown **Figure 3**.

(updated figures appear in Supplementary Fig. 16 in the Supplementary Information)

It would be beneficial to discuss how the efficiency is measured briefly. Simply saying that the cable lengths are different in the simulation and measurement may not be sufficient to explain the difference.

The antenna efficiency was measured using the MVG StarLab system. In the interest of brevity, we kindly refer the reviewer to the information regarding the StarLab system and the antenna measuring process, which can be found on their website: <https://www.mvg-world.com/en/products/antenna-measurement/multi-probe-systems/starlab>

(updated text appears between Line 519 and Line 520 in the manuscript)

Figure 3 | Corrected FPRFS antenna efficiency results. **a** Measured peak gain as a function of operating frequency for FPRFS emulated and conventional $1 \times 2-2-1$ patch. **b** Simulated efficiency, measured efficiency, and measured peak gain as a function of operating frequency for FPRFS emulated and conventional $2 \times 2-2-1$ patch. **c** Simulated efficiency, measured efficiency, and measured peak gain as a function of operating frequency for FPRFS emulated and conventional $3 \times 2-2-1$ patch.

2. The advantage of using the proposed approach to tune the frequency is fine, but it may also change the radiation characteristics, such as radiation pattern.

Yes, the reviewer's statement is correct. This involves the multi-objective optimization (MOO) design. Our current solution is to assign different weights to different parameters, which belong to the scaled multi-objective optimization (MOO) method¹. This method has three types of weights: equal weights, rank-order centroid weights, and rank-sum weights.

In our case, different weights are assigned depending on their priorities to achieve the optimal combination of SNR at receivers and VSWR at the transmitter. Different weights are assigned and sent to different receiving base stations to achieve a radiation pattern with the optimal SNRs at multiple base stations. Different weights are assigned to different target frequencies to shape the comprehensive VSWR curve at multiple frequencies of interest.

MOO problems are commonly faced in reconfigurable RF applications due to the large number of tuning parameters. Zapata Cano et al. (2023) presented an enhanced MOO algorithm offering a flexible design approach for electromechanically adaptable devices leveraging pixelated structures².

(updated text appears between Line 467 and Line 487 in the Supplementary Information)

3. The authors now also provided the total efficiency of the FPRFS. It would be good if the authors could briefly explain how the antenna efficiency is measured.

The antenna efficiency was measured using the MVG StarLab system. In the interest of brevity, we kindly refer the reviewer to the information regarding the StarLab system and the antenna measuring process, which can be found on their website: <https://www.mvg-world.com/en/products/antenna-measurement/multi-probe-systems/starlab>

(updated text appears between Line 519 and Line 520 in the manuscript)

Reviewer 2

Upon thorough review of the authors' response from the previous round of review and the revised manuscript, I am pleased to acknowledge the considerable efforts made by the authors in addressing the raised comments. The changes implemented have significantly enhanced and clarified the paper. However, before any potential recommendation can be made, there are still critical issues that necessitate consideration. As previously mentioned, the paper is well-organized and presents detailed results. However, one key aspect that lacks novelty is the limited exploration of new principles and mechanisms. The research content of this paper is more focused on technical issues, resulting in negligible innovation. Below are some suggestions for the authors to consider.

1. The authors have discussed the primary innovation achieved through the proposed FPRFS device in their current work. Nonetheless, in order to emphasize the enhancement and merit of the concept put forth in this study, it is imperative to undertake quantitative comparisons with prior research or state-of-the-art devices. This approach will enable a quantitative assessment of the advantages, such as the improved radiation efficiency/gain achieved by the proposed method. Such comparisons would be conducive to elucidating the authors' proposition.

The efficiencies achieved by reconfigurable antennas vary depending on their substrate, operating frequency, switching mechanism, RF switch characteristics, and reconfigurability of the antenna structure. Below is a comparison table.

Literature	Switching mechanism	Number of RF switches	Radiation efficiency	Other aspects
Feng et al.(2020) ³	Reflective metasurface	NA	41%-51%	Require other RF sources to function.
Jin et al.(2018) ⁴	PIN diode	4	<60%	Bandwidth: 2.25-3.16 GHz Total number of states: 4
Abutarboush et al.(2018) ⁵	PIN diode	2	40%-50%	Bandwidth: 1.5-4 GHz Total number of states: 4
Sheta et al.(2008) ⁶	PIN diode	3	20%-58%	Bandwidth: 620-1150 MHz Total number of states: 4
Hussain et al.(2015) ⁷	PIN diode	4	22%-40%	Bandwidth: 710-3600 MHz Total number of states: 4
This work	PIN diode	100	22%-43% (IP); 1.2%-1.8% (NIP)	Bandwidth: 850-3850 MHz Total number of states: 2 ⁶⁰

The reflective metasurface presented by Feng et al. is a special type of reconfigurable RF devices, which requires other RF sources to function. Except for reconfigurable metasurfaces, the antenna reconfigurability is limited in the other published work. We conjecture that the major reason for this loss is the growing number of RF switches. A key contribution of our work is that the limitation imposed by an increasing number of RF switches on compromised radiation efficiency is overcome, unlocking significantly increased potential in flexibility, programmability, and

scalability for reconfigurable antennas. Others' work achieved comparatively high radiation efficiency but their antenna reconfigurabilities are constrained to very few lossy RF switches. Although the current radiation efficiency achieved in our work is comparable to state-of-the-art reconfigurable antennas, an important point is that our efficiency does not degrade as the number of RF switches increases. This is novel and a significant contribution of our work. Our efficiency can be further increased by replacing the current FR4 substrate with a Rogers substrate with a lower dielectric constant and tangent loss and making the substrate thinner.

(updated text appears between Line 372 and Line 388 in the Supplementary Information)

2. The authors have clarified the optimization process in their response letter, highlighting it as a procedure of searching for optimal patterns and parameters within the parameter space. This method determines the optimal stimulation pattern by iteratively exploring all predetermined possible situations and selecting the one that enhances performance the most based on the signal received by the receiver, the base station in a conception. While this is a simple and direct method, it would be interesting to explore whether specific optimization algorithms, such as Artificial Neural Networks (ANN), can be implemented in this process.

Yes. Optimizing pixelated surfaces like our proposed FPRFS involves a binary multi-dimensional problem⁸. Artificial intelligence (AI) has been explored in various application areas for adaptive reconfigurable antenna arrays, including adaptive nulling, wireless localization, multiple-input multiple-output (MIMO) communications, element failures, and calibration⁹. The configuration of our pixelated FPRFS is solely determined by the ON/OFF patterns of all PIN diodes. Therefore, a single-bit binary array representing the ON/OFF states of all PIN diodes can serve as the optimization output (with '1' representing ON and '0' representing OFF). A neural network model can be developed to take desired RF characteristics, such as VSWR and radiation pattern, as input, and the optimized configuration as output. This model can be trained using a dataset containing FPRFS configurations and their corresponding RF characteristics.

Similar applications of optimizing reconfigurable RF devices using advanced optimization techniques have been widely explored with satisfactory results. Alkurt et al. (2021) presented a reconfigurable ground plane for a monopole antenna that can achieve the desired 3D antenna with the help of an ANN¹⁰. The reconfigurable ground plane consists of an 18×18 matrix of unit cells and each cell is represented by logic 1 or 0 for its states. The ANN is trained by 152 different configurations with simulated results, and takes S11 level, maximum radiation direction and gain as input parameters to generate the ground plane configuration as output. Montaser et al. (2022) built a Neural Network-based that can learn and predict the far-field radiation pattern of a designed metasurface with an array of reconfigurable unit cells¹¹. This method predicts the radiation pattern with high accuracy in a substantially shorter time period and can be integrated with adaptive reconfigurable RF designs effectively. Noh et al. (2022) presented a deep neural network (DNN) that can predict the radiation pattern of a reconfigurable metasurface and search for the best unit cell configuration for beam forming¹². This DNN-based approach is highly efficiency in managing

massive search volumes and is advantageous for complex active unit arrays like reconfigurable antennas. Kovaleva et al. (2020) presented a cross-entropy method to optimize the configuration of a metasurface for artificial magnetic conductor and phase shifter applications with desired characteristics⁸.

We acknowledge the importance of optimization and applicability of Neural Network techniques to address these problems. We have added a comment to refer an interested reader who has expertise in these techniques and may be able to apply them.

(updated text appears between Line 390 and Line 417 in the Supplementary Informaiton)

3. In the response regarding the self-adaptive capability, the authors seem to determine the optimization features, in turn, based on the far-field radiation characteristics with the highest SNR, as well as the impedance matching pattern that achieves the lowest VSWR. What they do is iteratively check each pattern in a loop to determine if it is an option with the highest SNR, and repeat the process to search for the pattern with the lowest VSWR. This raises the question of how the optimal combination of parameters can be obtained through sequential searching steps. In reality, this constitutes a multi-objective optimization design, therefore, it would be beneficial to explain how the device is optimized by employing different objective functions in the proposed processes.

Our current solution is to assign different weights to different parameters, which belong to the scaled multi-objective optimization (MOO) method¹. This method has three types of weights: equal weights, rank-order centroid weights, and rank-sum weights.

In our case, different weights are assigned depending on their priorities to achieve the optimal combination of SNR at receivers and VSWR at the transmitter. Different weights are assigned and sent to different receiving base stations to achieve a radiation pattern with the optimal SNRs at multiple base stations. Different weights are assigned to different target frequencies to shape the comprehensive VSWR curve at multiple frequencies of interest.

MOO problems are commonly faced in reconfigurable RF applications due to the large number of tuning parameters. Zapata Cano et al. (2023) presented an enhanced MOO algorithm offering a flexible design approach for electromechanically adaptable devices leveraging pixelated structures².

(updated text appears between Line 467 and Line 487 in the Supplementary Information)

4. In response to comment 6 from reviewer #3, the authors claim that the VNA in the current experimental setup can be replaced by the ADL 5960 in practical scenarios. While this may be true from a functionality viewpoint, the use of integrated vector network analyzers such as ADL 5960 in mobile devices may not be economically practical for consumer electronics, including smartphones, due to their high cost. It would be helpful if the authors could propose more practical solutions for such scenarios.

As discussed before, the VNA is not necessary for the self-optimization or the self-adaptation functionalities of the FPRFS and is only applied for experimental demonstration with spectrum convenience. An RF-DC converter would be suitable for estimating power, with faster response speed, cheaper cost, easier system integration, and better compatibility with signals with nonlinear frequency components. This can be physically implemented using a diode, a resistor and capacitor circuit, and an analog to digital converter.

The ADL 5960 comprises a resistive bidirectional bridge, downconversion mixers, intermediate frequency (IF) amplifiers and filters, and a local oscillator (LO). The bidirectional bridge/coupler, functioning signal separation for S-parameter measurements with high accuracy¹³, and the local oscillator, functioning IF signal generation for heterodyne detection, are the necessary parts for a VNA, while the rest are common components end applications. The ADL 5960 functions as a single on-chip VNA solution.

This system can be implemented using discrete components to greatly reduce costs for more economical and practical applications. For instance, a HHM2955A1 sub-miniature directional coupler (costs \$0.1) and an ADF4350 (costs \$10.7) can be used with a suitable RF front end.

Examples in the research literature of producing single-chip VNA chipsets include: Nasr et al. (2014) who presented a low-cost portable on-chip VNA system using the 0.35-um SiGe:C technology covering a frequency bandwidth between 50 and 100 GHz¹³. Nehring et al. (2016) presented a highly integrated VNA chipset operating between 4 and 32 GHz¹⁴. Chung et al. (2017) presented a single-chip SiGe reflectometer composed of a bridge coupler and two wideband heterodyne receivers, with a package size of 1.8 mm² and power consumption of 640 mW¹⁵. All these work would provide solutions for integrating our proposed FPRFS for more practical scenarios.

Reviewer 3

Authors replied to all the raised comments in the satisfactory manner.

References

1. Gunantara, N. A review of multi-objective optimization: Methods and its applications. *Cogent Eng.* 5, 1–16 (2018).
2. Zapata Cano, P. H. *et al.* Ultra-Low-loss Reconfigurable Phase-shifting Metasurface in V band: A Multi-objective Optimization Approach. *IEEE Trans. Antennas Propag.* 14, 1–1 (2023).
3. Feng, R. *et al.* Flexible Manipulation of Bessel-Like Beams with a Reconfigurable Metasurface. *Adv. Opt. Mater.* 8, 1–10 (2020).
4. Jin, G., Li, M., Liu, D. & Zeng, G. A Simple Planar Pattern-Reconfigurable Antenna Based on Arc Dipoles. *IEEE Antennas Wirel. Propag. Lett.* 17, 1664–1668 (2018).
5. Abutarboush, H. F. & Shamim, A. A Reconfigurable Inkjet-Printed Antenna on Paper Substrate for Wireless Applications. *IEEE Antennas Wirel. Propag. Lett.* 17, 1648–1651 (2018).
6. Sheta, A. F. & Mahmoud, S. F. A widely tunable compact patch antenna. *IEEE Antennas Wirel. Propag. Lett.* 7, 40–42 (2008).
7. Hussain, R. & Sharawi, M. S. Integrated reconfigurable multiple-input-multiple-output antenna system with an ultra-wideband sensing antenna for cognitive radio platforms. *IET Microwaves, Antennas Propag.* 9, 940–947 (2015).
8. Kovaleva, M., Bulger, D. & Esselle, K. P. Cross-Entropy Method for Design and Optimization of Pixelated Metasurfaces. *IEEE Access* 8, 224922–224931 (2020).
9. Zardi, F., Nayeri, P., Rocca, P. & Haupt, R. Artificial Intelligence for Adaptive and Reconfigurable Antenna Arrays: A Review. *IEEE Antennas Propag. Mag.* 63, 28–38 (2021).
10. Alkurt, F. O., Erkinay Ozdemir, M., Akgol, O. & Karaaslan, M. Ground plane design configuration estimation of 4.9 GHz reconfigurable monopole antenna for desired radiation features using artificial neural network. *Int. J. RF Microw. Comput. Eng.* 31, 1–12 (2021).
11. Montaser, A. M. & Mahmoud, K. R. Design of Intelligence Reflector Metasurface Using Deep Learning Neural Network for 6G Adaptive Beamforming. *IEEE Access* 10, 117900–117913 (2022).
12. Noh, J. *et al.* Reconfigurable reflective metasurface reinforced by

- optimizing mutual coupling based on a deep neural network. *Photonics Nanostructures - Fundam. Appl.* 52, (2022).
13. Nasr, I. *et al.* Single-and dual-port 50-100-GHz integrated vector network analyzers with on-chip dielectric sensors. *IEEE Trans. Microw. Theory Tech.* 62, 2168–2179 (2014).
 14. Nehring, J. *et al.* A 4-32-GHz Chipset for a Highly Integrated Heterodyne Two-Port Vector Network Analyzer. *IEEE Trans. Microw. Theory Tech.* 64, 892–905 (2016).
 15. Chung, H., Ma, Q., Sayginer, M. & Rebeiz, G. M. A 0.01-26 GHz single-chip SiGe reflectometer for two-port vector network analyzers. *IEEE MTT-S Int. Microw. Symp. Dig.* 1259–1261 (2017) doi:10.1109/MWSYM.2017.8058835.

REVIEWERS' COMMENTS

Reviewer #1 (Remarks to the Author):

The authors have addressed the comments made in the second review satisfactorily. This reviewer has no more comments.

Reviewer #2 (Remarks to the Author):

I am pleased to note that the authors have addressed the referees' comments from the previous two rounds of review, leading to a significant enhancement in the paper's quality. Consequently, I support the study for publication.